# Proton currents constrain structural models of voltage sensor activation

**Aaron L Randolph[1,2†], Younes Mokrab[3‡], Ashley L Bennett[1,2], Mark SP Sansom[3], Ian Scott Ramsey[1,2]***

[1]Department of Physiology and Biophysics, Virginia Commonwealth University School of Medicine, Richmond, United States; [2]Medical College of Virginia Campus, Virginia Commonwealth University School of Medicine, Richmond, United States; [3]Department of Biochemistry, University of Oxford, Oxford, United Kingdom

**Abstract** The Hv1 proton channel is evidently unique among voltage sensor domain proteins in mediating an intrinsic 'aqueous' $H^+$ conductance ($G_{AQ}$). Mutation of a highly conserved 'gating charge' residue in the S4 helix (R1H) confers a resting-state $H^+$ 'shuttle' conductance ($G_{SH}$) in VGCs and Ci VSP, and we now report that R1H is sufficient to reconstitute $G_{SH}$ in Hv1 without abrogating $G_{AQ}$. Second-site mutations in S3 (D185A/H) and S4 (N4R) experimentally separate $G_{SH}$ and $G_{AQ}$ gating, which report thermodynamically distinct initial and final steps, respectively, in the Hv1 activation pathway. The effects of Hv1 mutations on $G_{SH}$ and $G_{AQ}$ are used to constrain the positions of key side chains in resting- and activated-state VS model structures, providing new insights into the structural basis of VS activation and $H^+$ transfer mechanisms in Hv1.

*For correspondence: ian.
ramsey@vcuhealth.org

Present address: [†]Department of Anesthesiology, Yale University School of Medicine, New Haven, United States; [‡]Sidra Medical Research Center, Doha, Qatar

Competing interests: The authors declare that no competing interests exist.

## Introduction

The superfamily of voltage sensor (VS) domain proteins includes tetrameric voltage-gated cation channels (VGCs), voltage-sensitive phosphatases (VSPs) and the Hv1 proton channel. VS domains sense changes in membrane potential and undergo voltage-dependent conformational rearrangements that gate the ion channel and lipid phosphatase activities in associated effector domains. Hv1 lacks a separate effector domain, and instead mediates a depolarization-activated $H^+$-selective 'aqueous' conductance ($G_{AQ}$) that is intrinsic to the VS domain (*Ramsey et al., 2006, 2010; Sasaki et al., 2006*). Biophysical properties of $G_{AQ}$ gating in Hv1 are similar to pore domain gating in tetrameric VGCs (*Ramsey et al., 2006; Sasaki et al., 2006; Decoursey, 2003; Gonzalez et al., 2013*), and $G_{AQ}$ can therefore be used to directly monitor conformational changes in the Hv1 VS domain.

X-ray structures demonstrate that VS domains from phylogenetically distant species share a similar architecture: a membrane-integral bundle of four α-helices (S1-S4) surrounds an hourglass-shaped central crevice with hydrated vestibules facing the intra- or extra-milieux (*Long et al., 2005; Takeshita et al., 2014; Li et al., 2014; Long et al., 2007; Guo et al., 2016; Kintzer and Stroud, 2016; Zhang et al., 2012; Payandeh et al., 2011*). Hydrophobic groups appear to limit solvent accessibility at the waist of the central crevice in both resting- and activated-state VS structures, while ionizable side chains, including conserved Arg residues in S4, appear to be solvent-exposed (*Long et al., 2005; Takeshita et al., 2014; Li et al., 2014; Zhang et al., 2012; Payandeh et al., 2011; Krepkiy et al., 2009; Lacroix et al., 2014; Krepkiy et al., 2012*). Available VS domain structures are consistent with experimental data showing that the central crevice VS domain forms a pathway for the transmembrane movement of gating charge ($Q_G$) that is carried mainly by S4 Arg side chains (*Vargas et al., 2012; Bezanilla, 2008; Seoh et al., 1996; Aggarwal and MacKinnon, 1996*),

and that the transmembrane electrical field is focused within the central crevice (*Ahern and Horn, 2005*; *Starace and Bezanilla, 2004*; *Campos et al., 2007*).

Changes in membrane potential are thought to drive S4 to move from its resting 'down' conformation to its 'up' position in the activated VS, but estimates of the magnitude of vertical S4 displacement vary widely (from ~5 Å to ~20 Å), depending on the experimental technique used (*Li et al., 2014*; *Guo et al., 2016*; *Vargas et al., 2012*; *Posson et al., 2005*; *Ruta et al., 2005*; *Banerjee and MacKinnon, 2008*; *Larsson et al., 1996*; *Cha et al., 1999*). In contrast to S4, the S1-S3 helices appear to form a relatively immobile scaffold (*Long et al., 2005*; *Vargas et al., 2012*; *Ahern and Horn, 2005*; *Tombola et al., 2007*; *Jensen et al., 2012*; *Delemotte et al., 2011*). A highly conserved Phe residue in S2 ($F^{2.50}$; F150 in Hv1; refer to *Table 1* and *Figure 1—figure supplement 1* for the residue numbering scheme used here) faces into the VS central crevice and evidently participates in the formation of a hydrophobic barrier that helps to focus the electric field (*Ramsey et al., 2010*; *Long et al., 2007*; *Jensen et al., 2012*; *Wood et al., 2012*; *Freites et al., 2006*; *Gosselin-Badaroudine et al., 2012*). In the Shaker $K^+$ channel, $F^{2.50}$ (F290) exhibits state-dependent interactions with various S4 Arg side chains (*Lacroix et al., 2014*; *Tao et al., 2010*), and thereby serves as a spatial reference point in resting- and activated-state VS domain X-ray and model structures.

*In silico* studies of VS domain structure can help to bridge structural and experimental data by delineating probable atomic interactions, mapping solvent accessibility and identifying possible routes for ion conduction (*Ramsey et al., 2010*; *Jensen et al., 2012*; *Delemotte et al., 2011*; *Wood et al., 2012*; *Freites et al., 2006*; *Chamberlin et al., 2014, 2015*; *Kulleperuma et al., 2013*). Congruous with the X-ray structure of an Hv1-based chimeric protein (mHv1cc; pdb: 3WKV), Hv1 homology models generally agree that $F^{2.50}$ (F150) is appropriately located to participate in the

**Table 1.** Numbering of selected residues in Hv1 and Shaker VS domain sequences.

| Residue position | *Hs* Hv1 number | mHv1cc number | *Ci* Hv1 number | Shaker number |
|---|---|---|---|---|
| 1.48 | V109 | V105 | V157 | I237 |
| 1.51 | D112 | D108 | D160 | S240 |
| 1.52 | A113 | A109 | S161 | I241[*] |
| 1.54 | L115 | L111 | L163 | I243 |
| 1.55 | V116[†] | V112 | V164 | F244 |
| 2.44 | I144 | F140 | L192 | T284 |
| 2.46 | I146 | I142 | I194 | C286 |
| 2.47 | L147 | L143 | L195 | I287[*] |
| 2.50 | F150 | F146 | F198 | F290 |
| 2.51 | M151 | M147 | M199 | T291 |
| 3.53 | V177 | I173[‡] | V225 | A319 |
| 3.54 | V178 | V174 | V226 | I320 |
| 3.58 | F182 | F178 | F230 | F324 |
| 3.61 | D185 | D181 | D233 | L327 |
| 3.65 | L189 | L185 | I237 | V331 |
| 4.40 | L198 | L194 | I248 | M356 |
| 4.43 | L201 | L197 | L251 | L358 |
| 4.44 | I202 | I198 | V252 | A359 |
| 4.46 | L204 | L200 | L254 | L361 |
| 4.47 | R205 | R201 | R255 | R362 |

[*]Shaker I241H ($I^{1.52}$) and I287H ($I^{2.47}$) confer $G_{SH}$ (*Campos et al., 2007*).
[†]Functionally substitutes for $D^{1.51}$ in background of D112V (*Morgan et al., 2013*).
[‡]Ile$^{3.53}$ was transferred from *Ci* VSP into mHv1cc chimera (*Takeshita et al., 2014*).

formation of a hydrophobic barrier within the central crevice (*Ramsey et al., 2010*; *Wood et al., 2012*; *Chamberlin et al., 2014*, *2015*; *Kulleperuma et al., 2013*). Mutation of an acidic residue in S1 ($D^{1.51}$/D112) that is selectively conserved in Hv1 and VSPs causes large positive shifts in the $G_{AQ}$-V relation and compromises $H^+$ selectivity (*Ramsey et al., 2010*; *Musset et al., 2011*; *Berger and Isacoff, 2011*), consistent with its predicted location near $F^{2.50}$/F150 (*Ramsey et al., 2010*; *Wood et al., 2012*; *Chamberlin et al., 2014*, *2015*; *Kulleperuma et al., 2013*). Ionization of $D^{1.51}$/D112 was suggested to be necessary for $H^+$ transfer via $G_{AQ}$, but the necessity of an anion at this position to maintain exquisite $H^+$ selectivity suggests that the side chain is likely to remain ionized when $G_{AQ}$ is open (*Musset et al., 2011*). The permeability of $D^{1.51}$/D112 mutants, including D112H and D112K, to solution anions ($Cl^-$, $MeSO_3^-$ and possibly $OH^-$) strongly argues that the VS central crevice is well-hydrated in the Hv1 activated-state conformation, consistent with the hypothesis that $H^+$ conduction occurs in a water wire and does not require explicit ionization of protein side chains (*Ramsey et al., 2010*).

Although free energy changes calculated by a quantum mechanical (QM) model suggest that D112/$D^{1.51}$ can be neutralized (*Dudev et al., 2015*), the orientation of the two side chains contained in the simple model system used in this study (D112/$D^{1.51}$ and R2/$R^{4.50}$) differs substantially from that seen in activated-state Hv1 model structures (*Ramsey et al., 2010*; *Wood et al., 2012*; *Chamberlin et al., 2014*, *2015*; *Kulleperuma et al., 2013*). Computational approaches that explicitly define the proton hold promise for elucidating $H^+$ transfer mechanism(s), but their sensitivity to subtle geometric differences in various models reinforces the need for rigorous experimental testing of candidate model structures in advance of their implementation for calculating electronic structure. Experimental approaches that can be used to map the locations of functionally important residue side chains with high spatial resolution are therefore needed. Although $G_{AQ}$ measurement is a potentially powerful tool for exploring structure-function relationships in Hv1, the absence of $G_{AQ}$ in most VS domain proteins limits its more widespread implementation. Furthermore, experimental validation of electrically silent resting-state VS domain conformations, which may serve as useful controls for theoretical studies, is problematic.

Gain-of-function mutations are reported to confer resting-state proton-selective 'shuttle' ($G_{SH}$) or monovalent cation-nonselective 'omega' ($G_\Omega$) conductances in VS domain proteins, and residues that line the central crevice or 'gating pore' have been identified in several studies (*Starace and Bezanilla, 2004*; *Gosselin-Badaroudine et al., 2012*; *Tombola et al., 2005*; *Capes et al., 2012*; *Gamal El-Din et al., 2010*, *2014*; *Sokolov et al., 2005*; *Struyk and Cannon, 2007*). However, putative resting-state Hv1 VS domain X-ray and model structures contain hydrophobic (*Takeshita et al., 2014*; *Chamberlin et al., 2014*) or electrostatic (*Li et al., 2015*) barriers that would appear to prevent $G_{SH}$ and $G_\Omega$, consistent with the absence of resting-state currents in experimental studies in R1H (*Kulleperuma et al., 2013*) and R1A/C/S (*Ramsey et al., 2006*; *Sasaki et al., 2006*) mutant Hv1 channels, respectively. To address the paradoxical lack of $G_{SH}$ in Hv1 R1H (*Kulleperuma et al., 2013*), we expressed Hv1 R1H in mammalian cells and measured whole-cell currents under voltage clamp. We find that R1H does confer $G_{SH}$ in Hv1 without abrogating $G_{AQ}$. The effects of second-site mutations in the background of R1H impose tight spatial constraints on the positions of key residue side chains. We present new resting-state Hv1 VS domain model structures that are distinct from previous models and fully consistent with available experimental data.

## Results

The relative positions of conserved Arg residues in the S4 segments from *Drosophila* Kv1-family Shaker $K^+$ channel (*Dm* Shaker), voltage-sensitive phosphatase from *Ciona intestinalis (Ci* VSP), and human Hv1 (*Hs* Hv1) are shown in an amino acid sequence alignment (*Figure 1A*, *Figure 1—figure supplement 1*). To facilitate comparisons between disparate VS domain sequences and structures, we adopt a generic numbering system (*Table 1*, *Figure 1—figure supplement 1*) that is analogous to one used for G-protein coupled receptors (*Isberg et al., 2015*). In this numbering scheme, the most highly conserved S4 Arg residue, R2, is designated $R^{4.50}$ (*Figure 1A*, *Table 2*, *Figure 1—figure supplement 1*). Whereas the R1-R3 positions are conserved in most VS domain proteins, Hv1 uniquely bears a polar neutral residue (N214/N4/$N^{4.56}$) in the R4 position (*Figure 1A*, *Figure 1—figure supplement 1*).

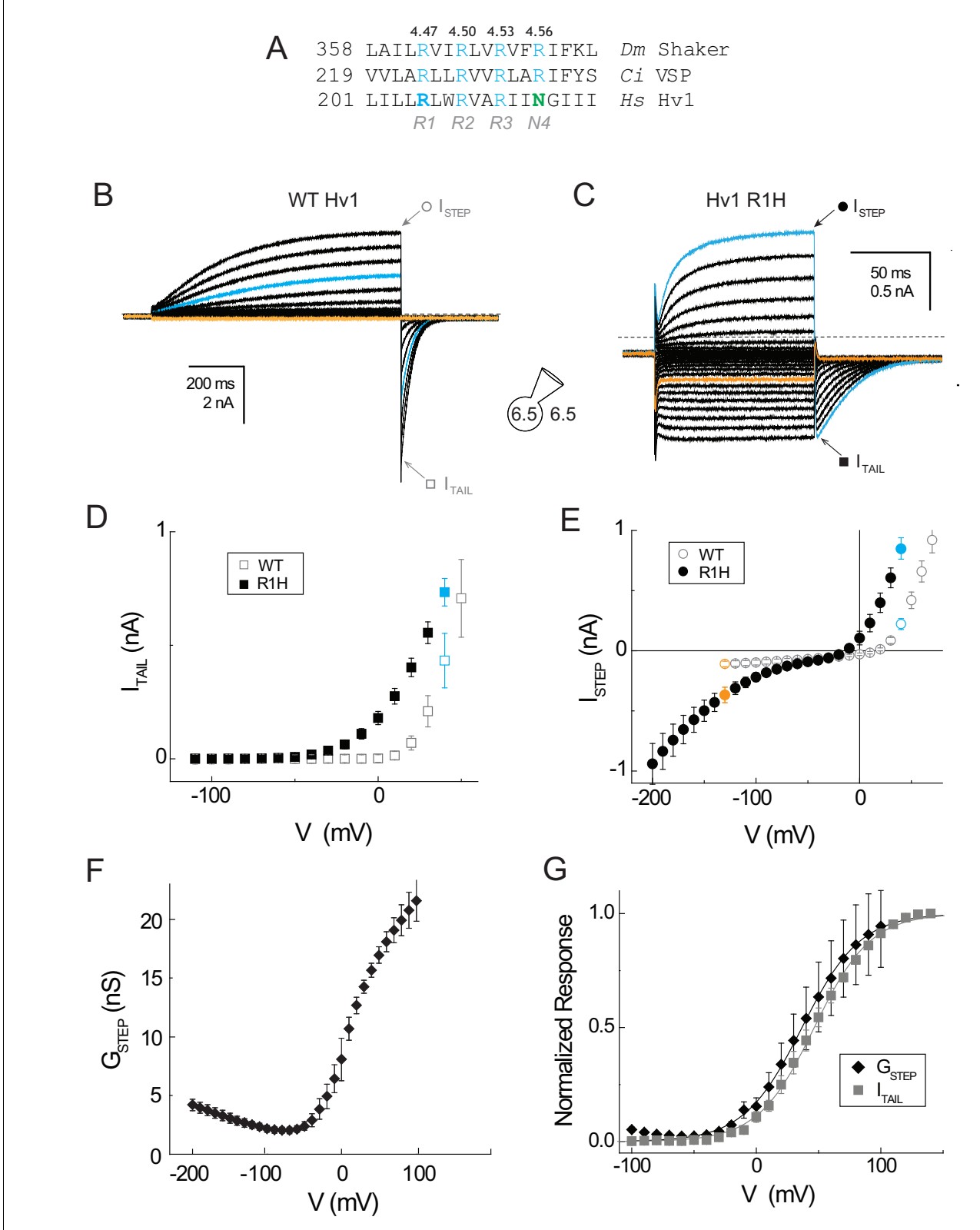

**Figure 1.** A resting-state H$^+$ 'shuttle' conductance in Hv1 R1H. (A) A multiple sequence alignment of the S4 helix in *Drosophila melanogaster* Shaker (GI:288442), *Ciona intestinalis* Voltage Sensitive Phosphatase (GI:76253898) and *Homo sapiens* Hv1 (GI:38783432) is shown. Numbering indicates amino acid position and bold typeface indicates residues that are mutated in this study. Conserved S4 Arg residues are shown in blue and Asn214 in Hv1 is green. (B, C) Whole-cell currents in a representative cell expressing Hv1 R1H are elicited by voltage steps from a holding potential of −60 mV to −130

*Figure 1 continued*

mV through +70 mV in increments of +10 mV in a representative cell expressing WT Hv1 (**B**) or from a holding potential of −50 mV to −200 mV through +40 mV in increments of +10 mV (**C**). Tail currents measured at −90 mV and recording solutions (pH$_O$ 6.5, pH$_I$ 6.5) are indicated in the diagram. Symbols indicate the approximate times at which I$_{STEP}$ (circles) and I$_{TAIL}$ (squares) are measured. Colored lines indicate currents measured at +40 mV (cyan) or −130 mV (orange) and dashed lines indicate the zero current amplitudes. (**D, E**) I$_{TAIL}$-V (**D**) and I$_{STEP}$-V (**E**) relations are shown for WT Hv1 (open symbols) and R1H (filled symbols). Colored symbols indicate currents measured at +40 mV (cyan) and −130 mV (orange), as shown in **B** and **C**. Symbols represent means ± SEM from *n* = 4 (I$_{STEP}$, WT), *n* = 6 (I$_{STEP}$, R1H), *n* = 6 (I$_{TAIL}$, WT), or *n* = 7 (I$_{TAIL}$, R1H) cells. Linear leak currents are subtracted from the I$_{TAIL}$-V relations in **D**. (**F**) The mean G$_{STEP}$-V relation calculated from data in E exhibits a 'U' shape in which the apparent maximal G$_{STEP}$ amplitudes at positive and negative potentials are unequal. The data suggest that G$_{AQ}$ is open at positive potentials while the resting-state H$^+$ shuttle conductance (G$_{SH}$) is open at more negative voltages. (**G**) The voltage dependence of the intrinsic activated-state H$^+$ conductance (G$_{AQ}$) in Hv1 R1H is estimated from I$_{TAIL}$ (gray squares) measured at −90 mV in symmetrical pH 6.5 recording solutions as shown in **B** and **C**. G$_{STEP}$ (black diamonds) is calculated from I$_{STEP}$ (see Materials and methods). G$_{STEP}$ and I$_{TAIL}$ are normalized to their apparent maxima in each cell and symbols represent the mean ± SEM from *n* = 6 (I$_{STEP}$) or *n* = 7 (I$_{TAIL}$) cells. Solid lines represent fits of the data between −50 mV and +100 mV to a Boltzmann function (I$_{TAIL}$, gray line: V$_{0.5}$ = 46.5 mV, dx = 22.6; G$_{STEP}$, black line; V$_{0.5}$ = 36.7 mV, dx = 23.4).

The following figure supplements are available for figure 1:

**Figure supplement 1.** Amino acid sequence alignments of S4 helical segments in VS domain proteins.

**Figure supplement 2.** Measurement of G$_{AQ}$ selectivity in Hv1 R1H.

His mutations at R$^{4.47}$ in Shaker (R362H) and *Ci* VSP (R221H) are each sufficient to confer a resting-state G$_{SH}$ (*Starace and Bezanilla, 2004*; *Villalba-Galea et al., 2013*), but resting-state current was not observed in Hv1 R205H (*Kulleperuma et al., 2013*). Cytotoxicity associated with the expression of a constitutive H$^+$ conductance can hamper efforts to measure G$_{SH}$ (*Campos et al., 2007*), so we expressed Hv1 R1H in a tetracycline-inducible congenic HEK-293 cells 1–4 days after induction (see Materials and methods). Consistent with previous reports (*Ramsey et al., 2010*; *Kulleperuma et al., 2013*), cells expressing EGFP-tagged WT Hv1 or Hv1 R1H display voltage- and time-dependent currents (*Figure 1B,C*). Current amplitudes during the voltage step (I$_{STEP}$) and immediately after subsequent hyperpolarization (I$_{TAIL}$) are typically larger in cells expressing WT Hv1 than in cells expressing R1H (*Figure 1B,C*). As previously reported (*Kulleperuma et al., 2013*), we find that the time courses of G$_{AQ}$ activation and deactivation are substantially more rapid in Hv1 R1H than WT Hv1 (*Figure 1B,C*). I$_{TAIL}$-V relations and V$_{THR}$ analyses demonstrate that G$_{AQ}$ activation is shifted negatively by −32 mV in R1H, from +7 mV in WT Hv1 (*Ramsey et al., 2010*) to −25.0 ± 1.9 mV (n = 20) in R1H (*Figure 1D*; *Table 2*). pH$_O$-dependent shifts in I$_{TAIL}$ reversal potentials (*Figure 1— figure supplement 2*; 52.2 mV/pH unit at pH$_I$ 6.5 and 50.5 mV/pH unit at pH$_I$ 7.0) are close to the

**Table 2.** Effects of Hv1 mutations on G$_{AQ}$ gating. I$_{TAIL}$ was measured in cells expressing the indicated constructs and V$_{THR}$ was estimated by visual inspection of raw current records as described (Materials and methods). The data represent means ± SEM from determinations in the indicated number (*n*) of cells.

| construct | G$_{AQ}$ V$_{THR}$ (mV) | SEM | *n* |
|---|---|---|---|
| WT* | +7 | 2 | 6 |
| N4R* | +17 | 3 | 4 |
| R1A* | +6 | 3 | 4 |
| R1H | −25.0 | 1.9 | 20 |
| R1H-N4R | −20.7 | 2.2 | 14 |
| D185A-R1H | +40.0 | 1.9 | 13 |
| D185H-R1H | +80.0 | 3.8 | 7 |

*\*Ramsey et al., 2010*.

Nernst prediction (58.2 mV/pH unit) under our recording conditions (~20°C), suggesting that R1H does not substantially alter $H^+$ selectivity for $G_{AQ}$.

## A resting-state '$H^+$ shuttle' conductance ($G_{SH}$) in Hv1 R1H

We routinely observe robust inward $I_{STEP}$ at negative potentials in cells expressing Hv1 R1H (*Figure 1C,E*). Whereas the $I_{STEP}$-V relation for WT Hv1 is outwardly rectifying, the $I_{STEP}$-V relation in R1H exhibits double rectification with an apparent plateau at intermediate voltages near −30 mV (*Figure 1E*). The inwardly-rectifying shape of the steady-state $I_{STEP}$-V relation in Hv1 R1H at negative potentials is similar to other R1H VS domain mutants (*Starace and Bezanilla, 2004*; *Capes et al., 2012*; *Struyk and Cannon, 2007*; *Villalba-Galea et al., 2013*) but distinct from the bell-shaped $I_{STEP}$-V relations in Shaker R2H and R3H that utilize a carrier-type ($G_{CA}$) mechanism for $H^+$ transfer (*Starace and Bezanilla, 2001*; *Starace et al., 1997*). To discriminate resting- and activated-state conductances in Hv1, we use $G_{SH}$ terminology in reference to the channel-like $H^+$ conductances observed in R1H VS domain mutants.

The $I_{STEP}$-V curve in Hv1 R1H exhibits prominent inward rectification at negative potentials (*Figure 1E*), similar to Shaker and *Ci* VSP R1H mutants (*Starace and Bezanilla, 2004*; *Villalba-Galea et al., 2013*), whereas at potentials > −30 mV, $I_{STEP}$-V curve in Hv1 R1H exhibits outward rectification like WT Hv1 (*Figure 1E*). The apparent 'plateau' in the $I_{STEP}$-V relation near −30 mV appears to result when both $G_{SH}$ and $G_{AQ}$ are close to their respective minima (*Figure 1E*). Consistent with this interpretation, the doubly-rectifying $I_{STEP}$-V relation gives rise to a 'U-shaped' $G_{STEP}$-V relation in Hv1 R1H (*Figure 1F*). The net $G_{STEP}$-V may be interpreted to represent the amalgam of distinct conductances ($G_{SH}$ and $G_{AQ}$) that have distinct voltage dependencies, opposite gating polarity and unequal maximal amplitudes. Classical ion channel gating theory predicts that $G = N \cdot \gamma \cdot P_{OPEN}$ (where $\gamma$ is unitary conductance, N is channel number and $P_{OPEN}$ is open probability), and we therefore infer that $G_{SH} = N_{SH} \cdot \gamma_{SH} \cdot P_{OPEN-SH}$ and $G_{AQ} = N_{AQ} \cdot \gamma_{AQ} \cdot P_{OPEN-AQ}$. If each Hv1 R1H mutant VS domain mediates both $G_{SH}$ and $G_{AQ}$ (albeit at different potentials), i.e., $N_{SH} = N_{AQ}$ and $\gamma_{AQ} \neq \gamma_{SH}$ (*Figure 1F*).

$I_{STEP}$-V and $I_{TAIL}$-V relations in WT Hv1 are apparently linear at negative voltages (*Figure 1D,E*), consistent with the expectation that $P_{OPEN-AQ}$ will asymptotically approach its minimum as the membrane potential becomes more negative (*Gonzalez et al., 2013*; *Cherny et al., 1995*). In R1H however, the inward $I_{STEP}$ clearly becomes larger as membrane potential becomes more negative (*Figure 1E*). $G_{STEP}$ also rises with additional hyperpolarization, suggesting that the voltage-dependent increase in inward current results from an increase in $P_{OPEN-SH}$ (*Figure 1F*). The notion that $G_{SH}$ gating reflects a change in VS conformation is consistent with results from a study conducted showing that *Ci* Hv1 exhibits kinetically distinct fluorescence changes with distinct voltage dependencies (*Qiu et al., 2013*). However, $G_{AQ}$ and $G_{SH}$ gating measured here are more widely separated than the fluorescence changes (*Qiu et al., 2013*), suggesting that $G_{SH}$ may report an earlier transition in the Hv1 activation pathway.

The $G_{STEP}$-V relation in Hv1 R1H exhibits a local minimum near −50 to −70 mV; at these intermediate potentials, $G_{STEP}$ could reflect contributions from $G_{AQ}$ and $G_{SH}$, in addition to voltage-independent membrane leakage ($G_{LEAK}$). Inspection of the $G_{AQ}$-mediated $I_{TAIL}$-V relation in R1H indicates that $P_{OPEN-AQ}$ is negligibly small at voltages negative to −40 mV (*Figure 1G*), but the unambiguous dissection of $G_{SH}$ gating is compromised by the contributions of $G_{SH}$ and $G_{LEAK}$ to the aggregate $G_{STEP}$. To measure $G_{SH}$ gating in isolation, we sought to test the hypothesis that mutagenesis could be used to experimentally block $G_{AQ}$. We therefore combined R1H with D112V, which is reported to abrogate $G_{AQ}$ (*Musset et al., 2011*), but so far we have been unable to measure either $G_{SH}$ or $G_{AQ}$ in cells expressing channels D112X-R1H double-mutant (where X is Val, Asn or Ala; not shown).

## N214R isolates the resting-state $G_{SH}$ from the intrinsic $G_{AQ}$

Previous studies show that N4R and N4K mutations attenuate outward $I_{STEP}$ mediated by $G_{AQ}$, but have comparatively little effect on inward $I_{TAIL}$, indicating that basic side chains at this position block the $H^+$ permeation pathway in a voltage-dependent fashion (*Ramsey et al., 2010*; *Sakata et al., 2010*). We therefore incorporated N4R into the background of Hv1 R1H (R1H-N4R) and measured expressed currents as described for the R1H single mutant. Hv1 R1H-N4R mediates robust inward currents carried by $G_{SH}$ like R1H, but outward $G_{AQ}$-mediated $I_{STEP}$ amplitude is substantially

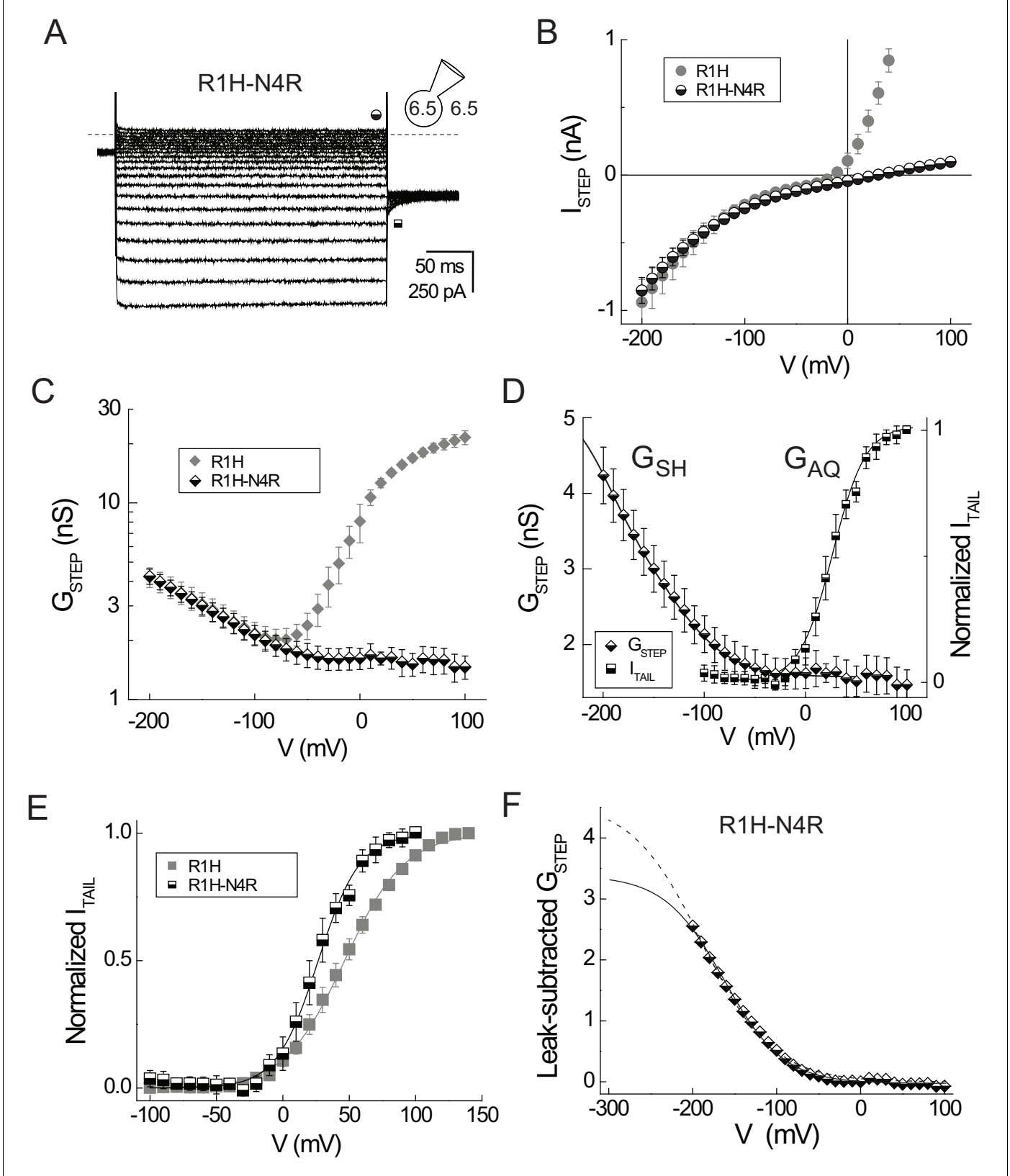

**Figure 2.** A second-site N4R mutation isolates $G_{SH}$ in Hv1 R1H. (**A**) Representative whole-cell currents in a cell expressing Hv1 R1H-N4R are elicited by voltage steps to −150 mV through +40 mV in increments of +10 mV from a holding potential of −30 mV; $I_{TAIL}$ is measured at −90 mV. Recording
*Figure 2 continued on next page*

*Figure 2 continued*

solutions are symmetrical pH 6.5, as indicated. Symbols indicate approximate times at which $I_{STEP}$ (half-filled circles) and $I_{TAIL}$ (half-filled squares) are measured; the dashed line indicates the zero current amplitude. (B) The $I_{STEP}$-V relations for R1H-N4R (half-filled circles) and R1H (filled gray circles; data from *Figure 1*) are plotted together for comparison. Note that outward $I_{STEP}$ mediated by $G_{AQ}$ is apparently absent in R1H-N4R but inward currents mediated by $G_{SH}$ are similar in R1H and R1H-N4R. For R1H-N4R, symbols represent means ± SEM $n$ = 4 cells. (C) $G_{STEP}$-V relations for R1H-N4R (half-filled diamonds) and R1H (filled gray diamonds; data from *Figure 1*) are shown without leak subtraction. Note the log scale for $G_{STEP}$. Data represent means ± SEM from $n$ = 3 cells expressing R1H-N4R that exhibited similar resting-state current amplitudes. (D) $G_{STEP}$-V (half-filled diamonds) and normalized $I_{TAIL}$-V (half-filled squares) relations for R1H-N4R are scaled to illustrate their relative positions on the voltage axis. Data represent means ± SEM from $n$ = 3 cells with similar current amplitudes. $I_{TAIL}$ data from each cell is linear leak-subtracted and normalized to its amplitude at +100 mV prior to averaging. Lines represent fits of the data to single Boltzmann functions ($G_{STEP}$: $G_{STEPmin}$ = 1.5 nS, $G_{STEPmax}$ = 5.5 nS, $V_{0.5}$ = −172.1 mV, dx = 40.4; $I_{TAIL}$: $V_{0.5}$ = 26.3 mV, dx = 16.3). (E) Leak-subtracted normalized $I_{TAIL}$-V relations for R1H (filled gray squares; data from *Figure 1*) and R1H-N4R (half-filled squares) are plotted together for comparison. Lines represent fits to single Boltzmann functions (R1H: $V_{0.5}$ = 46.5 mV, dx = 22.6; R1H-N4R: $V_{0.5}$ = 26.3 mV, dx = 16.3). (F) $G_{SH}$ gating was estimated by fitting the leak-subtracted $G_{STEP}$-V relation to Boltzmann functions in which $V_{0.5}$ is either free to vary (solid line: $G_{SHmax}$ = 3.4 nS, $V_{0.5}$ = −164 mV, dx = 35.4) or constant (see *Figure 4F*, dashed line: $V_{0.5}$ = $V_{PEAK}$ = −189 mV, $G_{SHmax}$ = 4.6 nS, dx = 42.1).

The following figure supplement is available for figure 2:

**Figure supplement 1.** Estimating $G_{AQ}$ gating parameters from $G_{STEP}$-V relations in Hv1 R1H.

reduced, and the remaining current exhibit a linear dependence on membrane potential and is thus attributable to $G_{LEAK}$ (*Figure 2A,B*). The time course of $I_{TAIL}$ decay in N4R (*Ramsey et al., 2010*) and R1H-N4R (*Figure 2A*) is evidently monophasic and notably lacks the 'hook' seen in the presence of the gating modifier 2GBI (*Hong et al., 2013*), indicating that R1H-N4R channels are open, but blocked, at positive voltages, and relief of a block occurs instantaneously upon hyperpolarization (*Figure 2A*).

Unlike R1H, the steady-state R1H-N4R $I_{STEP}$-V relation is inwardly rectifying (*Figure 2B*). A comparison of R1H and R1H-N4R $G_{STEP}$-V relations shows that the $G_{AQ}$ component is absent in R1H-N4R, and $G_{SH}$ approaches saturable minimum at voltages positive to ~−30 mV (*Figure 2C*). Linear subtraction of the leakage ($G_{LEAK}$ = 1.5 ± 0.2 nS at +100 mV and $G_{LEAK}$ = 1.6 ± 0.2 nS at 0 mV; n = 4 cells) yields a $G_{STEP}$-V relation that is readily fit to a single Boltzmann function (*Figure 2F*), although ambiguity about the maximal amplitude of $G_{SH}$ at large negative potentials does not permit unambiguous determination of the $V_{0.5}$ or slope factors determined from curve fitting (*Figure 2F*). Although $G_{AQ}$ is blocked at positive potentials in R1H-N4R, the inward $I_{TAIL}$ carried by $G_{AQ}$ remains measurable, and $V_{THR}$ for activation of $G_{AQ}$ is similar in R1H and R1H-N4R (*Figure 2A,D*; *Table 2*), Boltzmann fits of the respective $I_{TAIL}$-V relations illustrate that the fitted slope value is steeper and midpoint ($V_{0.5}$) is ~20 mV more negative in R1H-N4R compared to R1H (*Figure 2E*). Although N4R dramatically decreases outward current carried by $G_{AQ}$, the second-site mutation appears to have only a modest effect on $G_{AQ}$ gating. Wide separation in the positions of the normalized G-V relations (*Figure 2D*) indicates that $G_{SH}$ and $G_{AQ}$ gating report thermodynamically distinct steps in the Hv1 activation pathway.

Our data suggest that voltage-dependent closure of $G_{SH}$ reports initial VS activation while $G_{AQ}$ gating reflects a late gating transition. Our results are similar, but not identical, to Shaker R1H (*Starace and Bezanilla, 2004*). For example, the time courses of $G_{SH}$ opening and closing in Hv1 R1H-N4R (*Figure 2A*) are evidently faster than Shaker and *Ci* VSP R1H mutants (*Starace and Bezanilla, 2004*; *Villalba-Galea et al., 2013*), possibly indicating that $G_{SH}$ gating in Hv1 does not require substantial conformational rearrangement of the protein backbone. In contrast, the time course of $G_{AQ}$-mediated $I_{STEP}$ and $I_{TAIL}$ (*Figures 1C*, *2A*) are comparatively slow, suggesting that activation and deactivation gating requires more extensive protein conformational rearrangements. $G_{SH}$ gating and $G_{AQ}$ block by N4R are likely to involve rapid local changes in the orientation of side chains that lie in or near the focused electrical field, and $G_{SH}$ gating phenomenologically resembles pore block by a permeant ion. $G_{\Omega}$ gating in R1A/C/S mutant Shaker channels is similarly attributed to block by the protein-associated side chains of S4 Arg residues, which permeate the 'gating pore' during VS activation. Consistent with the small apparent gating valence (0.5–0.7 $e_0$; *Figure 2F*) estimated from Boltzmann fits, local reorientation of the imidazole side chain in the introduced His at R1 could account for the voltage dependence of $G_{SH}$ gating.

## D185 selectively stabilizes the activated, $G_{AQ}$-open conformation of the Hv1 VS domain

Differences in the voltage dependence of $G_{SH}$ and $G_{AQ}$ gating suggest that mutation of residues which selectively stabilize the activated-state Hv1 VS conformation may preferentially perturb $G_{AQ}$ gating. An acidic residue in S3, D185/D$^{3.61}$, is conserved only in Hv1 (*Figure 1—figure supplement 1*). D185 mutations produce dramatic shifts in $V_{THR}$ toward positive potentials (*Ramsey et al., 2010*) without altering H$^+$ selectivity (*Musset et al., 2011*), consistent with the hypothesis that this residue participates in an interaction that stabilizes the $G_{AQ}$-open, activated-state conformation. We therefore introduced D185A and D185H mutations into the background of R1H and measured their effects on $G_{SH}$ and $G_{AQ}$ gating. As in Hv1 R1H and R1H-N4R, cells expressing D185-R1H double mutants manifest robust steady-state inward currents at negative membrane potentials (*Figure 3A, B*; *Figure 3—figure supplement 1*). As expected, the $G_{AQ}$ gating is shifted positively in D185H-R1H and D185A-R1H (*Figure 3C,D*; *Figure 3—figure supplement 1A,B*). Compared to R1H, $V_{THR}$ is shifted +65 mV in D185A-R1H and +105 mV in D185H-R1H (*Table 2*; *Figure 3G,H*); the effects of D185 mutations in the background of R1H are similar to the effects of single D185A or D185H mutations (*Ramsey et al., 2010*).

Like R1H alone, D185H-R1H and D185A-R1H exhibit U-shaped $G_{STEP}$-V relations (*Figure 3D*; *Figure 3—figure supplement 1A*) that are similar to R1H (*Figure 2*), indicating that $G_{SH}$ is not abrogated by D185 mutation. In contrast to R1H, the $I_{TAIL}$-V and $G_{STEP}$-V relations in D185A-R1H and D185H-R1H exhibit a wider plateau at intermediate potentials (*Figure 3D*; *Figure 3—figure supplement 1A*), which facilitates $G_{LEAK}$ subtraction (*Figures 3D*, *Figure 3—figure supplement 1*) and analysis of $G_{SH}$ gating. At negative voltages where $G_{SH}$ is open, the D185H-R1H and D185A-R1H $I_{STEP}$-V and $G_{STEP}$-V relations are similar to R1H and R1H-N4R (*Figure 3D*; *Figure 3—figure supplement 1A,B*), indicating that $G_{SH}$ gating is unaffected by D185 mutation. D185 mutations therefore appear to selectively destabilize the $G_{AQ}$-open conformation of the Hv1 VS domain, but do not alter interactions that are important for resting-state stabilization.

## First derivative analyses of G-V relations

Next we sought to test the hypothesis that changes in $G_{SH}$ gating can also be experimentally measured. However, the lack of $G_{SH}$ saturation at negative potentials limits our ability to accurately determine $G_{SH}$ gating parameters for from fits of $G_{STEP}$-V data to a Boltzmann function, even when the contributions to the net $G_{STEP}$ from $G_{SH}$ and $G_{LEAK}$ are defined (*Figure 2F*). To circumvent this limitation, we reasoned that an analysis of the first derivatives of $G_{STEP}$-V relations ($dG_{STEP}/dV$) could be a useful alternative approach. First, we simulated ideal $G_{STEP}$-V relations using a Boltzmann function (*Figure 3—figure supplement 2A,C*). As expected, a plot of $dG_{STEP}/dV$ vs. the applied potential ($dG_{STEP}/dV$-V) yields a bell-shaped distribution (*Figure 3—figure supplement 2B,D*). Fitting the data to a Gaussian function allows us to estimate the voltage at which the curve peaks ($V_{PEAK}$). Fitted $V_{PEAK}$ values are correlated to $V_{0.5}$ in simulated $G_{SH}$ and $G_{AQ}$ Boltzmann distributions (*Figure 3—figure supplement 2E*). Altering the amplitude and position of simulated $G_{STEP}$-V relations to reflect the known effect of changing pH$_O$ on $G_{AQ}$ gating produces a commensurate shift (40 mV/pH unit) in $V_{PEAK}$ (*Figure 3—figure supplement 2A–E*). We confirm that experimentally-measured $V_{0.5}$ (estimated by Boltzmann fitting of $I_{TAIL}$-V relations) and $V_{PEAK}$ (from Gaussian fits to $dI_{TAIL}/dV$-V) values are similarly pH$_O$-dependent using experimental $I_{TAIL}$ data in R1H-N4R (*Figure 3—figure supplement 2F,G*). The slopes of the $V_{0.5}$-pH$_O$ and $V_{PEAK}$-pH$_O$ relations in R1H-N4R are each close to 40 mV/pH unit (*Figure 3—figure supplement 2G*).

Our analyses of simulated and experimental data indicate that $V_{PEAK}$ can be used to estimate the positions of G-V relations when experimental conditions preclude direct measurement of either $G_{min}$ or $G_{max}$. We therefore compared estimated $G_{AQ}$ gating parameters in R1H, D185A-R1H and D185H-R1H determined from analyses of $V_{PEAK}$ and $V_{THR}$. In D185H-R1H, $I_{TAIL}$ does not clearly reach saturation at voltages ≤+200 mV, but the $dG_{STEP}/dV$-V relation rises to a peak near +150 mV and falls again at more positive potentials (*Figure 3E*). Although we did not measure R1H-D185A over as wide a range of positive potentials, we observe a peak in the $dG_{STEP}/dV$-V data near +100 mV (*Figure 3E*), suggesting that the midpoint of the $G_{AQ}$-V relation was reached. Gaussian fits of data from R1H, D185A-R1H and D185H-R1H yield $V_{PEAK}$ values of +23.3 mV, +98.9 mV and +144.3 mV, respectively (*Figure 3E*), and $V_{PEAK}$ is well-correlated to $V_{THR}$ (*Figure 3G*).

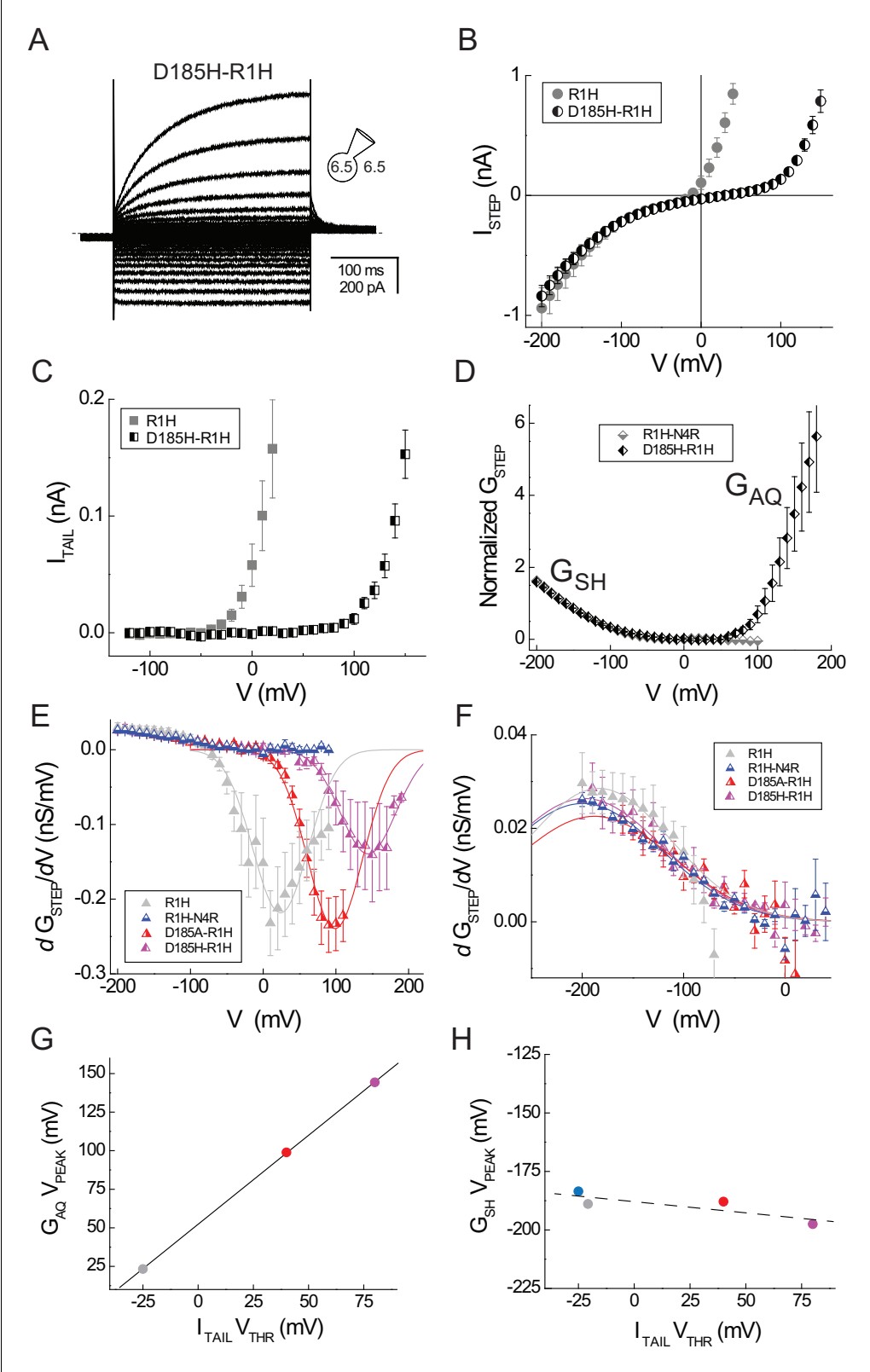

**Figure 3.** D185 mutations selectively affect $G_{AQ}$ gating. (**A**) Representative whole-cell current records elicited by voltage steps from −120 mV to +150 mV in a cell expressing Hv1 D185H-R1H in symmetrical pH 6.5 recording solutions are shown. (**B**) The $I_{STEP}$-V relation for D185H-R1H (half-filled circles) is compared to R1H (filled gray circles; data from *Figure 1*). (**C**) The mean $I_{TAIL}$-V relation for D185H-R1H (half-filled squares) is compared to R1H (filled gray squares; data from *Figure 1*). Linear leak currents are subtracted from the data. (**D**) The normalized leak-subtracted $G_{STEP}$-V relations for D185H-

*Figure 3 continued*

R1H (black half-filled diamonds) and R1H-N4R (gray half-filled diamonds; data from *Figure 2*) are compared. Linear $G_{LEAK}$ calculated between 0 mV and +50 mV is subtracted from the D185H-N4R data and $G_{STEP}$ is normalized to its value at −140 mV. A Boltzmann fit to the mean D185H-R1H $G_{STEP}$-V relation between −200 mV and +50 mV ($G_{MAX}$ = 2.2, dx = 35.3, $V_{0.5}$ = −164.9 mV; not shown) yields similar gating parameters to R1H-N4R (see *Figure 2*). (E) $dG_{STEP}/dV$-V relations are calculated from leak-subtracted $G_{STEP}$-V data measured in cells expressing R1H (gray triangles), R1H-N4R (blue triangles), D185A-R1H (red triangles), or D185H-R1H (violet triangles). For clarity, only data between −200 mV and +40 mV are shown in panel **F**. Lines represent Gaussian fits to the data between −100 mV and +100 mV (R1H, gray line: A = −20.8, ω = 76.2, $V_{PEAK}$ = +23.3 mV), −20 mV and +110 mV (D185A-R1H, red line: A = −22.9, ω = 76.2, $V_{PEAK}$ = +98.9 mV), or 0 mV and +190 mV (D185H-R1H, violet line: A = −13.3, ω = 76.2, $V_{PEAK}$ = +144.3 mV). For Gaussian fits to R1H and D185A-R1H data, ω is constrained to the value determined from a fit to D185H-R1H data (ω = 76.2). D185A-R1H and D185H-R1H data represent means ± SEM from $n$ = 3 cells; R1H data are replotted from *Figure 2* and R1H-N4R data is replotted from *Figure 4*. (F) Symbols represent $dG_{STEP}/dV$-V relations (panel **E**) between −200 mV and +40 mV only and lines represent Gaussian fits to the data (R1H, gray line: A = 5.1, ω = 143.8, $V_{PEAK}$ = −183.5 mV; R1H-N4R, blue line: A = 5.6, ω = 155.5, $V_{PEAK}$ = −188.9 mV; D185A-R1H, red line: A = 4.4, ω = 155.5, $V_{PEAK}$ = −187.9 mV; D185H-R1H, violet line: A = 5.1, ω = 155.5, $V_{PEAK}$ = −197.5 mV). (G, H) $V_{PEAK}$ values for $G_{AQ}$ (from **E**) and $G_{SH}$ (from **F**) gating are plotted against $V_{THR}$ for $G_{AQ}$-mediated $I_{TAIL}$ (data from *Table 2*) in cells expressing R1H (gray circle), R1H-N4R (blue circle), D185A-R1H (red circle) and D185H-N4R (violet circle). Note that $G_{AQ}$ $V_{PEAK}$ is not measured for R1H-N4R (**G**). Effects of D185 mutations on $G_{AQ}$ gating estimated from $V_{PEAK}$ and $V_{THR}$ are strongly correlated (solid black line in **G**, R = 0.99) whereas the effects of mutations on $G_{SH}$ gating are weakly correlated with their effects $G_{AQ}$ gating (dashed black line in **H**, R = −0.84).

The following figure supplements are available for figure 3:

**Figure supplement 1.** Effects of D185 mutations on $G_{AQ}$ and $G_{SH}$ gating in R1H.

**Figure supplement 2.** Estimating $G_{AQ}$ and $G_{SH}$ gating parameters from the first derivative of $G_{STEP}$-V.

In contrast to $G_{AQ}$ gating, $dG_{STEP}/dV$-V relations at negative voltages are similar in R1H, R1H-N4R, D185A-R1H and D185H-R1H (*Figure 3F*), and the fitted $V_{PEAK}$ values indicate that $G_{SH}$ gating is poorly correlated with the $V_{THR}$ for $G_{AQ}$ gating (*Figure 3H*). First derivative analyses of G-V relations therefore appear to quantitatively agree with results obtained using the established $V_{THR}$ method (*Musset et al., 2008*). We noted earlier that the apparent maximal amplitudes of $G_{AQ}$ and $G_{SH}$ ($G_{AQmax}$ and $G_{SHmax}$, respectively) are distinct (*Figures 1F*, *2C*), but our estimate of $G_{SHmax}$ remains tentative (*Figure 2F*). Using $V_{PEAK}$ determined from first derivative analysis ($V_{PEAK}$ = −189 mV = $V_{0.5}$) to constrain Boltzmann fits to the R1H-N4R data yields a revised estimate of $G_{SHmax}$ and the slope factor for $G_{SH}$ gating ($G_{SHmax}$ = 4.6 nS, dx = 42.1; *Figure 2F*, dashed line).

By subtracting the voltage-independent leak ($G_{LEAK}$ = 1.5 nS in R1H-N4R; *Figure 2C*), we calculated the net $G_{STEP}$-V for R1H (*Figure 1F*) and estimated the voltage dependence of $G_{AQ}$ gating ($V_{0.5}$ = 29.4 mV; *Figure 2—figure supplement 1*), which compares favorably with the value determined from direct measurement of the R1H-N4R $I_{TAIL}$-V relation ($V_{0.5}$ = 26.3 mV; *Figure 2E*). The foregoing analysis allows us to directly compare $G_{AQmax}$ (22.2 nS; *Figure 2—figure supplement 1*) and $G_{SH\ max}$ (4.6 nS; *Figure 2F*); after leak subtraction, $G_{AQmax}/G_{SHmax}$ = 4.8. Assuming that the maximum open probabilities for $G_{AQ}$ and $G_{SH}$ ($P_{OPENmax-AQ}$ and $P_{OPENmax-SH}$) are equal, the data suggest that the respective unitary conductances ($\gamma_{AQmax}$ and $\gamma_{SHmax}$, respectively) also differ by a factor of ~5. Stated differently, the data indicate that the *capacity* for $H^+$ transfer via the His-dependent $G_{SH}$ pathway is about 5 times smaller than that of the intrinsic $G_{AQ}$.

## $G_{SH}$ and $G_{AQ}$ gating are equally sensitive to changes in extracellular pH

A hallmark feature of $G_{AQ}$ gating in native and expressed Hv1 channels is the sensitivity of $G_{AQ}$ gating to changes in the pH gradient (*Ramsey et al., 2006*; *Sasaki et al., 2006*; *Cherny et al., 1995*). Mutations of candidate ionizable residues surprisingly failed to alter the sensitivity to changes in $pH_O$ (*Ramsey et al., 2010*), and the molecular mechanism for △pH sensing remains unknown. A kinetic model of Hv1 gating predicts that a voltage-independent transition governs $G_{AQ}$ opening, and this gating step could also be required for the channel's strong sensitivity to changes in △pH (*Villalba-Galea, 2014*). In order to determine whether earlysteps in the Hv1 activation pathway are sensitive to changes in △pH, we measured $G_{SH}$ gating in cells expressing Hv1 R1H-N4R at $pH_O$ 5.5, 6.5 and 7.5 (*Figure 4A–C*). Consistent with the effect of extracellular acidification to increase the driving force for inward $H^+$ current, $I_{STEP}$ increases as $pH_O$ is lowered (*Figure 4D*). The $I_{STEP}$-V

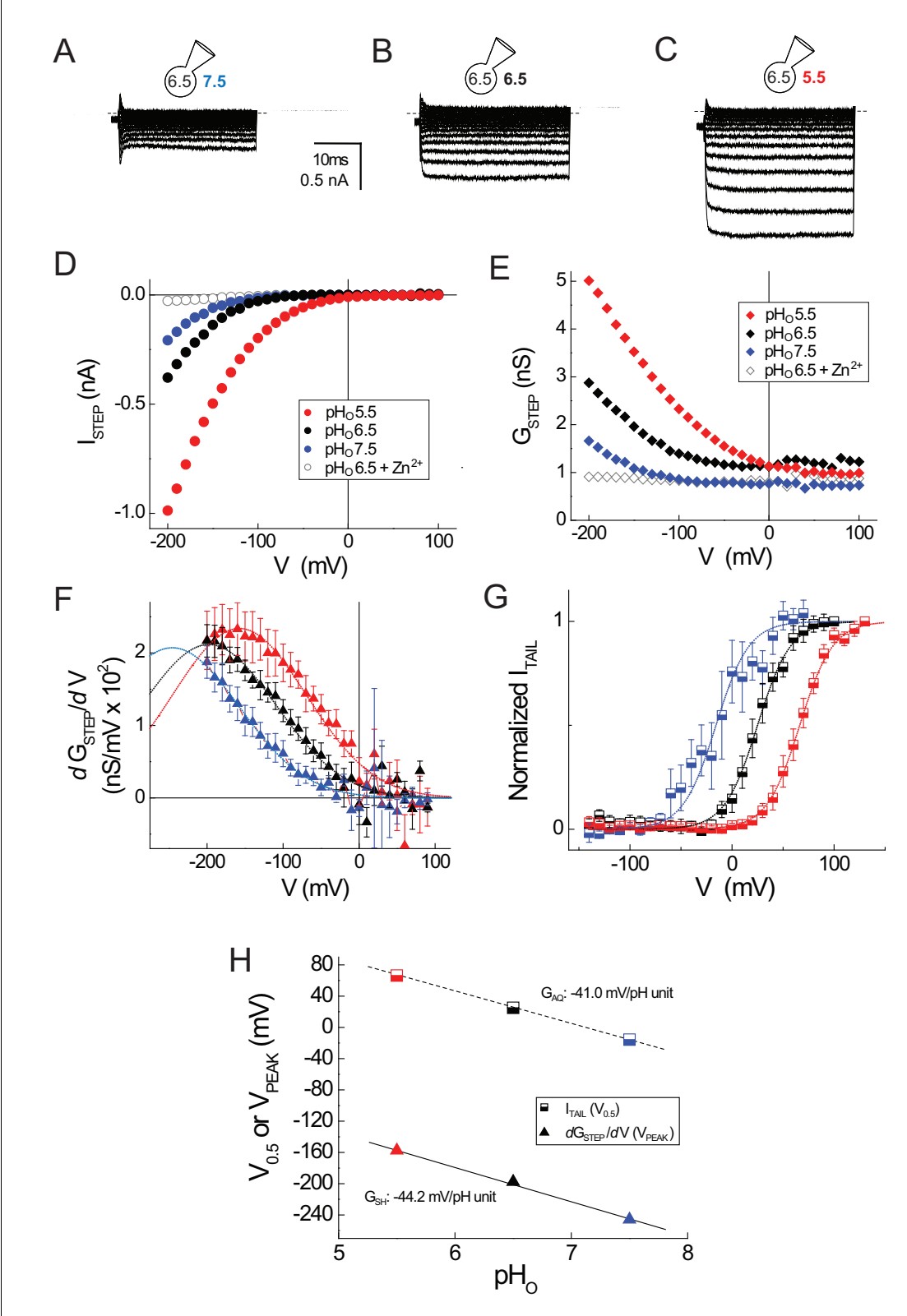

**Figure 4.** $G_{SH}$ and $G_{AQ}$ gating in R1H-N4R are similarly sensitive to changes in $pH_O$. (A–C) Representative whole-cell currents elicited by voltage steps (−200 mV to +100 mV in 10 mV increments) in a cell expressing R1H-N4R that was superfused with $pH_O$ 7.5 (A), $pH_O$ 6.5 (B) and $pH_O$ 5.5 (C) recording solutions are shown. (D) $I_{STEP}$-V relations at $pH_O$ 7.5 (filled blue circles), $pH_O$ 6.5 (filled black circles) or $pH_O$ 5.5 (filled red circles) are shown for the records in A–C. Open black circles represent $I_{STEP}$ measured at $pH_O$ 6.5 + 1 mM $Zn^{2+}$ in the same cell (raw traces not shown). (E) $G_{STEP}$-V relations at

*Figure 4 continued on next page*

Figure 4 continued

pH$_O$ 7.5 (filled blue diamonds), pH$_O$ 6.5 (filled black diamonds), pH$_O$ 5.5 (filled red diamonds) or pH$_O$ 6.5 + 1 mM Zn$^{2+}$ (open black diamonds) calculated from the data in D are shown. (F) $dG_{STEP}/dV$ is plotted in function of the membrane potential at which G$_{STEP}$ was measured at pH$_O$ 5.5 (red triangles), pH$_O$ 6.5 (black triangles) or pH$_O$ 7.5 (blue triangles) in R1H-N4R. Data points represent means ± S.E.M. from n = 8 (pH$_O$ 5.5), n = 12 (pH$_O$ 6.5) or n = 10 (pH$_O$ 7.5) cells. Dashed lines represent fits of the mean $dG_{STEP}/dV$-V relations between −200 mV and +100 mV to Gaussian functions: red line, pH$_O$ 5.5: A = 5.2, ω = 176.4, V$_{PEAK}$ = −157.4 mV; black line, pH$_O$ 6.5: A = 4.7, ω = 176.4, V$_{PEAK}$ = −197.6 mV; blue line, pH$_O$ 7.5: A = 4.6, ω = 176.4, V$_{PEAK}$ = −245.7 mV. (G) I$_{TAIL}$-V relations after steps to the indicated voltages at pH$_O$ 7.5 (half-filled blue squares), pH$_O$ 6.5 (half-filled black squares), or pH$_O$ 5.5 (half-filled red squares) are shown. I$_{TAIL}$ is normalized to the apparent maximum current at each pH$_O$. Data points represent means ± S.E.M. in n = 9 (pH$_O$ 5.5), n = 10 (pH$_O$ 6.5) or n = 4 (pH$_O$ 7.5) cells. Lines represent fits of the mean data to Boltzmann functions (red line, pH$_O$ 5.5: V$_{0.5}$ = +66.3 mV, dx = 16.2; black line, pH$_O$ 6.5: V$_{0.5}$ = +25.0 mV, dx = 16.2; blue line, pH$_O$ 7.5: V$_{0.5}$ = −15.7 mV, dx = 16.2). (H) The pH$_O$ dependence of G$_{SH}$ (estimated from fitted V$_{PEAK}$ values in mean $dG_{STEP}/dV$-V relations; triangles) and G$_{AQ}$ (estimated from fitted V$_{0.5}$ values in mean I$_{TAIL}$-V relations; squares) is compared. Lines represent linear fits of the data (G$_{SH}$: −44.2 mV/pH unit; G$_{AQ}$: −41.0 mV/pH unit).

relations remain inwardly rectifying for each pH tested, suggesting that G$_{AQ}$ block by N4R is not perturbed by changing pH$_O$ (*Figure 4D*).

Congruent with the effect of changing pH$_O$ on I$_{STEP}$, G$_{STEP}$ amplitude also varies with pH$_O$ in R1H-N4R (*Figure 4E*). To determine whether changing pH$_O$ shifts the apparent position of the G$_{SH}$-V relation, we compared the $dG_{STEP}/dV$-V relations at pH$_O$ 5.5, 6.5 and 7.5 (*Figure 4F*). Gaussian fits to the data reveal that V$_{PEAK}$ (pH$_O$7.5, V$_{PEAK}$ = −227 ± 9 mV, *n* = 11; pH$_O$ 6.5, V$_{PEAK}$ = −180 ± 7 mV, *n* = 14; pH$_O$ 5.5, V$_{PEAK}$ = −156 ± 10 mV, *n* = 8) is sensitive to changes in pH$_O$ (*Figure 4F*). To directly compare the pH$_O$ dependence of G$_{AQ}$ and G$_{SH}$ gating in R1H-N4R, we also measured I$_{TAIL}$ at pH$_O$ 5.5, 6.5 and 7.5 and fit the normalized data to a Boltzmann function (*Figure 4G*). Similar to WT Hv1, V$_{0.5}$ (G$_{AQ}$ gating) shifts −41.0 mV/pH unit in R1H-N4R (*Figure 4H*). Interestingly, V$_{PEAK}$ (G$_{SH}$ gating) shifts −44.2 mV/pH unit (*Figure 4H*), indicating that G$_{SH}$ and G$_{AQ}$ gating are similarly sensitive to changes in pH$_O$. Together with the effect of D185 mutations on G$_{AQ}$ gating, our findings imply that △pH-dependent gating occurs early in the Hv1 activation pathway, and later steps (like G$_{AQ}$ opening) inherit their △pH sensitivity from a previous gating transition. The sensitivity of G$_{SH}$-V relations to changes in pH$_O$, but not to D185 mutation, further reinforces our conclusion that G$_{AQ}$ and G$_{SH}$ report thermodynamically distinct gating transitions.

## An experimentally-constrained model of the Hv1 VS domain resting-state structure

R1H mutations are sufficient to confer phenomenologically similar G$_{SH}$ in VS domains from Hv1, Shaker and *Ci* VSP, suggesting the mechanism of H$^+$ transfer and resting-state VS structure are similar. A likely mechanism is H$^+$ shuttling mechanism via ionizable nitrogen atom(s) in the imidazole ring of the introduced His (*Starace and Bezanilla, 2004*). H$^+$ delivery to and removal from the introduced His presumably requires that hydrogen bonds are formed between nitrogen atoms and intra- and extra-cellular waters, and protons short-circuit the sharply-focused electrical field as they are shuttled by the introduced His (*Starace and Bezanilla, 2004*). The introduced His imidazole ring side chain is therefore likely to be in or near the hydrophobic barrier, and thus close to F150/F$^{2.50}$, in the G$_{SH}$-open, resting-state VS domain conformation. With the exception of *At* TPC1 DII VS domain X-ray structures (*Guo et al., 2016*; *Kintzer and Stroud, 2016*), R1/R$^{4.47}$ is not close to F$^{2.50}$ in putative resting-state VS domain X-ray structures, and the structural basis for H$^+$ transfer via G$_{SH}$ in R1H mutants remains unclear (*Takeshita et al., 2014*; *Li et al., 2014*; *Vargas et al., 2012*; *Jensen et al., 2012*; *Delemotte et al., 2011*; *Chamberlin et al., 2014*; *Li et al., 2015*).

We therefore generated a new resting-state Hv1 VS domain model (Hv1 D) in which R1 is located adjacent to F150/F$^{2.50}$ and subjected the Hv1 D model to all-atom molecular dynamics (MD) simulations as described previously (*Ramsey et al., 2010*). We also subsequently produced an R1H mutant resting-state Hv1 model structure (Hv1 E) and subjected the mutant model to MD simulation. The backbone structures of Hv1 D and the recently-solved X-ray structures of the domain II VS from *At* TPC1 (*At* TPC1 DII VS; pdb: 5W1J and 5DQQ), which adopts a resting-state conformation (*Guo et al., 2016*; *Kintzer and Stroud, 2016*), are remarkably similar (*Figure 5F*). The main difference between Hv1 D and *At* TPC1 DII VS domains is the tilt of S4 relative to membrane normal. S4 is more vertically oriented in Hv1 D than *At* TPC1 DII (*Figure 5F*), but given that S4 is likely to be

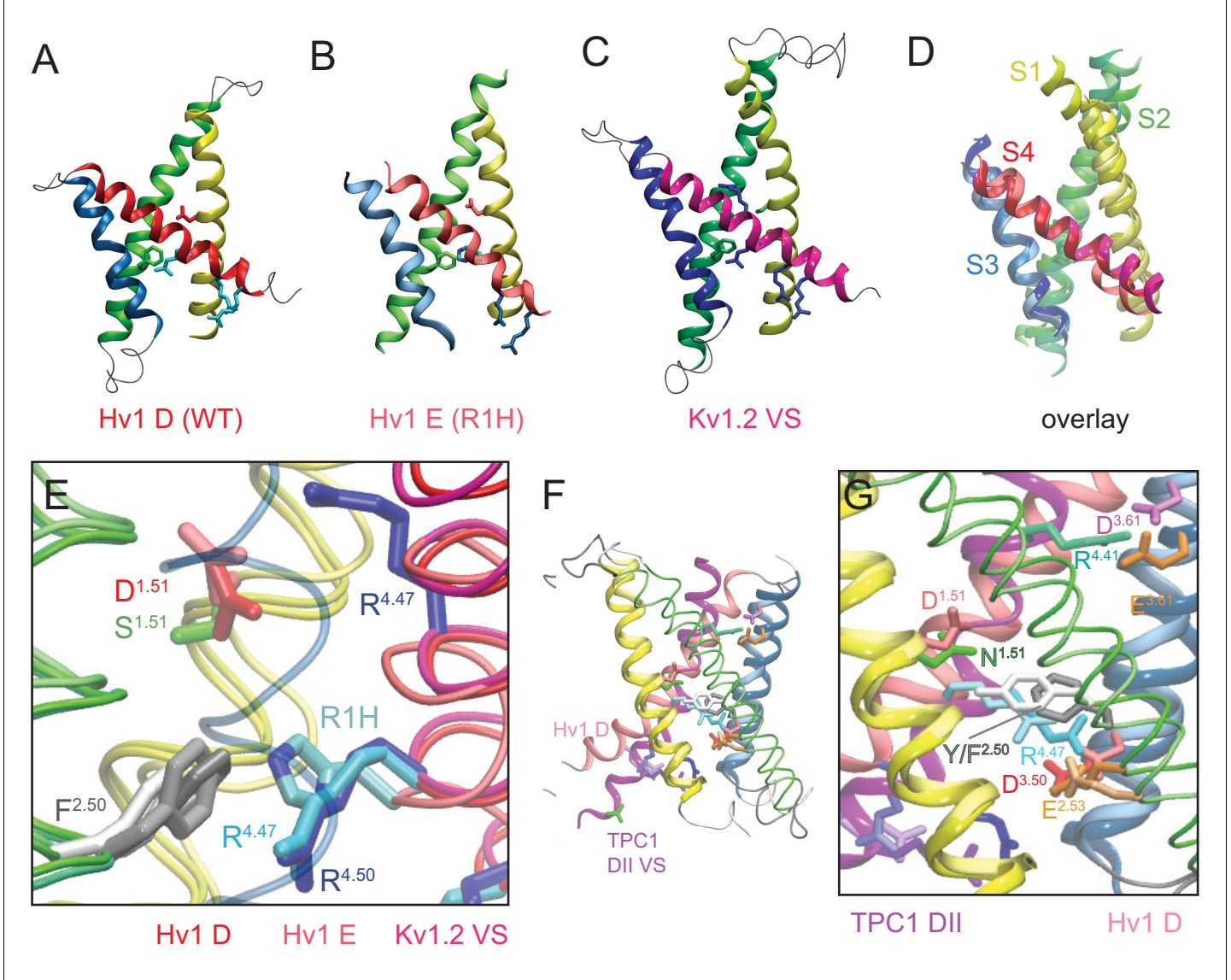

**Figure 5.** New Hv1 D (WT) and Hv1 E (R1H) VS domain resting-state model structures. (A–D) Ribbon diagrams represent backbone structures in snapshots taken from MD simulations of the Hv1 D (WT) VS domain resting-state model structure (A), Hv1 E R1H mutant model structure (B), resting-state Kv1.2 VS domain Rosetta model structure (*Pathak et al., 2007*) that was used as the template for construction of Hv1 D (C). The model structures in A–C are overlain in D to illustrate their overall structural similarity. Transmembrane helical backbones in A–D are color coded: S1, yellow; S2, green; S3, blue; S4 red. *Video 1* shows similar representations of Hv1 D, Hv1 E and Kv1.2 resting-state model structures rotated about the vertical axis. (E) An overlay of the Hv1 D, Hv1 E and Kv1.2 resting-state Rosetta VS domain model structures illustrates the relative positions of S1-S4 helical backbones (tubes colored as in A–D). Selected side chains (Hv1 D: D112/$D^{1.51}$, red; F150/$F^{2.50}$, light gray; R1/R205/$R^{4.47}$, cyan; Hv1 E: D112/$D^{1.51}$, light red; F150/$F^{2.50}$, white; R1H/R205H, cyan/blue; Kv 1.2: S176/$S^{1.51}$, green; F233/$F^{2.50}$, dark gray; R1/$R^{4.47}$ and R2/$R^{4.50}$, blue) are shown in colored licorice. For clarity, only the S3 helix from Hv1 D (transparent blue tube) is shown. (F) The backbone structures of Hv1 D model and *At* TPC1 DII X-ray (pdb: 5EJ1) VS domains are overlain. S1, S3 and S4 helices are shown as ribbons and S2 helices are shown as tubes. Helical segments are colored as in A–D and loop regions are gray; lighter shades represent Hv1 D and darker shades represent TPC1. Selected side chains in Hv1 D/TPC1 (D/$N^{1.51}$, red/green; F/$Y^{2.50}$, gray/white; D/$E^{3.61}$, magenta/orange) are shown in colored licorice. (G) A magnified view of the overlain Hv1 D and TPC1 structures illustrates the similar positions of selected side chains, which are shown in colored licorice (Hv1 D: D112/$D^{1.51}$, pale red; F150/$F^{2.50}$, light gray; E153/$E^{2.53}$, pale orange; D174/$D^{3.50}$, pale red; D185/$D^{3.61}$, pale magenta; R1/R205/$R^{4.47}$, pale cyan; TPC1: N443/$N^{1.51}$, green; Y475/$Y^{2.50}$, white; E478/$E^{2.53}$, orange; D500/$D^{3.50}$, red; E511/$E^{3.61}$, orange; R531/$R^{4.41}$, aqua; R537/R1/$R^{4.47}$, cyan).

The following figure supplements are available for figure 5:

**Figure supplement 1.** Comparison of Hv1 D to Kv1.2–2.1 chimera model and *Ci* VSP 'down' X-ray VS domain resting-state structures.

*Figure 5 continued on next page*

Figure 5 continued

**Figure supplement 2.** Comparison of Hv1 D and Hv1 E to Kv1.2 resting-state Rosetta model, *Ci* Hv1 resting-state model, and mHv1cc closed-state X-ray VS domain structures.
**Figure supplement 3.** Atomic distances and central crevice hydration in Hv1 D and Hv1 E MD simulations.
**Figure supplement 4.** Structure of the resting-state $G_{SH}$ permeation pathway.

highly mobile, the subtle difference in S4 tilt is perhaps not surprising. As suggested by protein sequence alignment (*Figure 1—figure supplement 1*) R537, rather than R531 (*Guo et al., 2016*; *Kintzer and Stroud, 2016*), in *At* TPC1 DII VS occupies a similar position as R1/R205/R$^{4.47}$ in Hv1 (*Figure 5G*), and we therefore define R537 as R1/R$^{4.47}$. R1/R$^{4.47}$ $C_\alpha$ positions in Hv1D and *At*TPC1 DII are separated by 2.9 Å, and the side chains of these residues are similarly directed to the intracellular side of F$^{2.50}$ (*Figure 5G*). Small differences in $C_\alpha$ distances are also measured between D/N$^{1.51}$ (1.0 Å), F$^{2.50}$ (2.0 Å) and D/E$^{3.61}$ (3.5 Å) in Hv1 D/*At* TPC1 DII VS, and these side chains are also oriented similarly in both structures (*Figure 5G*). In summary, the structural similarity between the *At* TPC1 DII X-ray structure (*Guo et al., 2016*; *Kintzer and Stroud, 2016*) and Hv1 D model VS domains strongly argues that our *Table 1* new Hv1 model represents a thermodynamically stable protein conformation.

As in *At* TPC1 DII VS X-ray structures, we find that the R1/R$^{4.47}$ terminal amine is oriented toward the intracellular vestibule in Hv1 D, where it is predicted to participate in a Coulombic interaction with a conserved acidic residue, D174/D$^{3.50}$ (*Figure 5G*; *Figure 5—figure supplement 4F*), that is part of the intracellular electrostatic network (*Ramsey et al., 2010*; *Long et al., 2005*). In Hv1 E, D174/D$^{3.50}$ interacts primarily with R2/R$^{4.50}$, rather than R1H (*Figure 5—figure supplement 1G*), which is consistent with our observation that the $G_{AQ}$-V relation is slightly shifted toward negative potentials. The Coulombic interaction between R1/R$^{4.47}$ and D174/D$^{3.50}$ appears to help to stabilize a $G_{AQ}$-closed, VS resting-state conformation. R1/R$^{4.47}$ also forms a salt bridge with D112/D$^{1.51}$ in Hv1 D (*Figure 5—figure supplement 1F*), but experimental data show that the main effect of D112 mutations is to shift the $G_{AQ}$-V relation toward positive potentials, suggesting that D112 plays a more important role in activated-state stabilization than resting-state stabilization (*Ramsey et al., 2010*). Consistent with this interpretation, we find that although D112 makes a stable electrostatic interaction with a protonated nitrogen atom of the R1H imidazole ring in Hv1 E, R1H only moderately shifts the $G_{AQ}$-V relation (*Table 2*; *Figure 1G*). Later we explore possible activated-state interactions between D112 and R3/R$^{4.53}$ (Figure 6A). Although Coulombic interactions involving D112 are reorganized in Hv1 E compared to Hv1 D (*Figure 5—figure supplement 1F,G*), distances between $C_\alpha$ atoms of selected atoms (D112/D$^{1.51}$, F150/F$^{2.50}$, D185/D$^{3.61}$ and R1/R205/R$^{4.47}$) are nonetheless similar (*Figure 5—figure supplement 3A,B*), illustrating that the VS domain architecture is similarly stable in Hv1 D and Hv1 E models.

Comparisons of available resting-state model and X-ray VS domain structures suggest an emerging pattern: the vertical position of S4 relative to S1-S3 is characteristically different in Hv1 D and *At* TPC1 DII compared to other VS domain X-ray structures. For example, the register of S4 Arg residues is shifted by one helical turn in the Kv1.2 resting-state model, where R2 (R$^{4.50}$) occupies the same position as R1/R$^{4.47}$ in Hv1 D (*Figure 5E*; *Video 1*). Similar differences in the register of S4 Arg residues are noted when Hv1 D is compared to putative resting-state conformations in Kv1.2–2.1 chimera (Kv chimera) VS domain models (*Jensen et al., 2012*; *Delemotte et al., 2011*) or the *Ci* VSD 'down' (*Ci* VSD$_D$, pdb: 4G80) (*Li et al., 2014*) X-ray structure (*Figure 5—figure supplement 1*). In Hv1 FL, the R1/R$^{4.47}$ side chain extends into the extracellular vestibule and the R2 side chain is close to F$^{2.50}$, similar to Kv 1.2, Kv chimera and *Ci* VSD$_D$ (*Figure 5—figure supplement 1E*; *Video 1*). Despite divergent approaches used to elucidate possible resting-state structures, the backbone structure and positions of other key residues in *Ci* VSD$_D$, including D$^{1.51}$/D129 and F161/F$^{2.50}$ are nearly superimposable with their positions in Hv1 D (*Figure 5—figure supplement 1D,E*). As expected, the backbone structures of S1-S4 helices in Hv1 D (WT) and Hv1 E (R1H) are also quite similar to the Kv1.2 resting-state (*Pathak et al., 2007*) template structure (*Figure 5A–E*; *Video 1*).

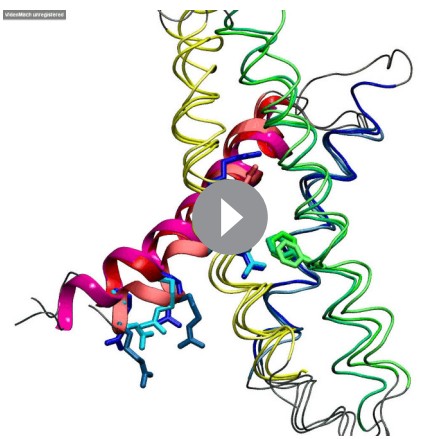

**Video 1.** Rotating side view of superimposed Kv1.2 resting-state, Hv1 D and Hv1 E model structures. Protein backbone and side chains are as described in *Figure 5A–D*. The animation shows the structures in rotation about the vertical axis.

Consistent with the experimental observation that R1H confers $G_{SH}$, and R1H is therefore readily accessible to intra- and extra-cellular solvent (*Starace and Bezanilla, 2004*), we observe that R1H is accessible to water molecules from both sides of the membrane, and the central crevice is similarly hydrated during Hv1 D and Hv1 E MD simulations (*Figure 5—figure supplement 3C–I*). The imidazole group of R1H is adjacent to F150/F$^{2.50}$, midway between D112/D$^{1.51}$ and D174/D$^{3.50}$, and appears to be appropriately positioned to shuttle protons between waters in the intracellular and extracellular vestibules (*Figure 5E*; *Figure 5—figure supplement 3C–I*; *Videos 2*, *3*). In contrast, the central crevices in a *Ci* Hv1 VS domain resting-state model structure (*Chamberlin et al., 2014*) and the mHv1cc Hv1/VSP/GCN4 chimeric protein (mHv1cc; pdb: 3WKV) are occupied by hydrophobic side chains (*Figure 5—figure supplement 4A–C*; *Videos 2*, *3*). In mHv1cc, a cluster of aliphatic side chains caps the central crevice on the extracellular side

of R1 (*Figure 5—figure supplement 4A*; *Videos 2*, *3*), preventing formation of a continuous hydrated pathway for H$^+$ transfer (*Takeshita et al., 2014*), and extracellular water access to R1/R$^{4.47}$ is also evidently precluded in *Ci* Hv1 (*Figure 5—figure supplement 4B*; *Videos 2*, *3*). Although hydrophobic side chains form a ring that surrounds the central hydrophilic crevice in Hv1 D and Hv1 E, R1 and R1H side chains are clearly visible within the gating pores when the model structures are viewed from the extracellular space (*Figure 5—figure supplement 4C*; *Video 3*).

In both Hv1 D and Hv1 E, we find that D112/D$^{1.51}$ is on the extracellular side of F$^{2.50}$ and is readily accessible to solvent (*Figure 5E*; *Figure 5—figure supplement 3C–I*). The position of D112/D$^{1.51}$ is consistent with experimental data showing that D112 is required for exquisite H$^+$ selectivity via $G_{AQ}$ and mutant channels (other than D112V or D112E, which are either non-functional or similar to WT Hv1, respectively) are permeable to anions (*Musset et al., 2011*), strongly arguing that the environment around D$^{1.51}$ is solvent-exposed. Consistent with experimental data showing that V116 (V$^{1.55}$) is functionally redundant with D$^{1.51}$ in supporting $G_{AQ}$ (*Morgan et al., 2013*), we find that V$^{1.55}$ is physically close to D$^{1.51}$ in Hv1 D and Hv1 E models (*Figure 5—figure supplement 4C*). However, we have so far been unable to measure currents associated with $G_{SH}$ (or $G_{AQ}$) in HEK-293 cells

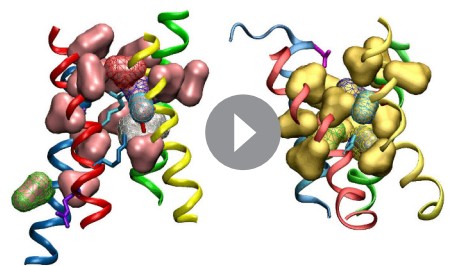

**Video 2.** Rotating side view of Hv1 E and mHv1cc. Protein backbone and side chains are as described in *Figure 5—figure supplement 4A–C*. Animation shows the structures in rotation about the vertical axis.

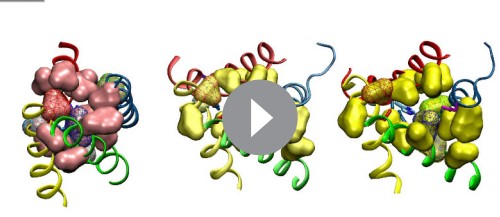

**Video 3.** Rocking extracellular view of mHv1cc, Hv1 D and Hv1 E. Protein backbone and side chains are as described in *Figure 5—figure supplement 4A–C*. The animation shows the structures viewed from the extracellular side in a rocking motion.

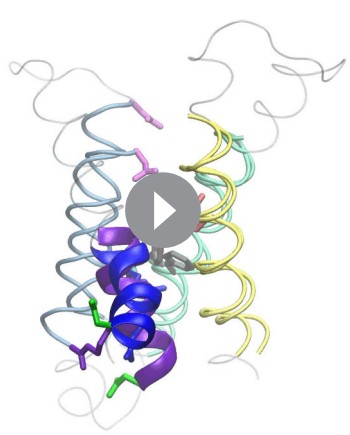

**Video 4.** Rotating view of Hv1 D and Hv1 FL resting-state model structures. S1-S3 helices are represented by colored tubes (S1, yellow; S2, green; S3, blue), S4 segments are shown as thick colored ribbons (Hv1 D, blue; Hv1 FL, violet) and loops (for Hv1 D only) are shown as thin gray tubes. Selected side chains are shown in colored licorice (D112, red; F150, gray; D185, magenta; R1, cyan; R2, blue; R3, violet; N4, green). The animation shows the structures in rotation about the vertical axis.

expressing R1H-D112A, R1H-D112N or R1H-D112V double-mutant channels (not shown), and it remains unclear whether R1H-D112X mutations disrupt the structure of the permeation pathway, displace necessary water molecules, or attenuate plasma membrane targeting. In contrast to Hv1 D and Hv1 E, the $D^{1.51}$, R1 and R2 side chains are closely packed into a hydrophobic crevice (*Figure 5—figure supplements 4A*, *Videos 2*, *3*) and evidently shielded from waters in mHv1cc (*Takeshita et al., 2014*).

$D185/D^{3.61}$ mutations dramatically shift the $G_{AQ}$-V relation toward positive potentials, indicating that this acidic side chain is likely to participate in an interaction that stabilizes the activated-state VS conformation. Interestingly, the Hv1 D185-equivalent ($E511/E^{3.61}$) is also present in the *At* TPC1 DII VS domain (*Figure 1—figure supplement 1*), but in the X-ray structures $E511/E^{3.61}$ interacts with $R531/R^{4.41}$ (*Guo et al., 2016*; *Kintzer and Stroud, 2016*). $R531/R^{4.41}$ is not conserved in Hv1 (*Figure 1—figure supplement 1*), suggesting that $D^{3.61}$ has a specific function in Hv1 that is not shared among other VS domains. Identifying the interacting partner(s) of $D185/D^{3.61}$ in Hv1 model structures is therefore of interest. In Hv1 FL, $D185/D^{3.61}$ is close enough to engage in a Coulombic interaction with R1 and appears to be solvent-accessible (*Figure 5—figure supplement 4F*; *Video 4*), but in mHv1cc, hydrophobic side chains fill the extracellular vestibule and the $D^{3.61}$/D181 carboxylate is tightly packed between non-polar side chains, including $F178/F^{3.58}$, $L184/L^{3.64}$, $L197/L^{4.43}$ and $L200/L^{4.46}$ (*Figure 5—figure supplement 2D,F*; *4A*; *Figure 6—figure supplement 4F*), and evidently inaccessible to solvent (*Takeshita et al., 2014*). Consistent with experimental data showing that D185 mutations do not alter $G_{SH}$ gating, we observe that D185 is located at the extracellular end of S3 in a solvent-accessible location that is distant from R1 in Hv1 D (*Figure 5—figure supplements 3H*, *4F*).

In contrast to the Hv1 D resting-state model, $D185/D^{3.61}$ is located close to $R3/R^{4.53}$ in the activated-state Hv1 B model (*Ramsey et al., 2010*). We show here that introduction of the N214R (N4R) mutation into Hv1 B does not alter the overall structure or stability of the VS domain, and $D185/D^{3.61}$ remains close enough to R3 to participate in a stable Coulombic interaction with $R3/R^{4.53}$ (*Figure 6*; *Figure 6—figure supplement 1*). As observed previously in Hv1 B (*Ramsey et al., 2010*), $R3/R^{4.53}$ makes a bidentate interaction with $D112/D^{1.51}$ and $D185/D^{3.61}$ in Hv1 B N4R (*Figure 6*; *Figure 6—figure supplement 1*). However, N4R addition allows $D112/D^{1.51}$ to form a new salt bridge with N4R (*Figure 6—figure supplement 1*) that may help stabilize the $G_{AQ}$-open conformation; this arrangement is in good agreement with experimental effects of N4R and N4K mutations, which markedly slow the timecourse of $I_{TAIL}$ decay (*Ramsey et al., 2010*). Although we have not explicitly tested this hypothesis using computational approaches, the N4R side chain appears to be appropriately positioned to sense changes in the electrical field that is thought to be focused near $F150/F^{2.50}$ (*Ahern and Horn, 2005*; *Starace and Bezanilla, 2004*). Rapid movement of a cationic N4R terminal amine within the electrical field is consistent with the experimental observation that outward currents carried by $G_{AQ}$ exhibit rapid voltage-dependent block/unblock in Hv1 R1H-N4R (*Figure 2*).

The availability of experimentally-refined resting- and activated-state Hv1 model structures suggests that the models could provide insights into the conformational changes associated with VS activation. Consistent with a generally accepted model of VS activation (*Vargas et al., 2012*), we find that the main difference between our experimentally-constrained activated- and resting-state Hv1 VS domain model structures is the position of S4 relative to the S1-S3 bundle, which appears to

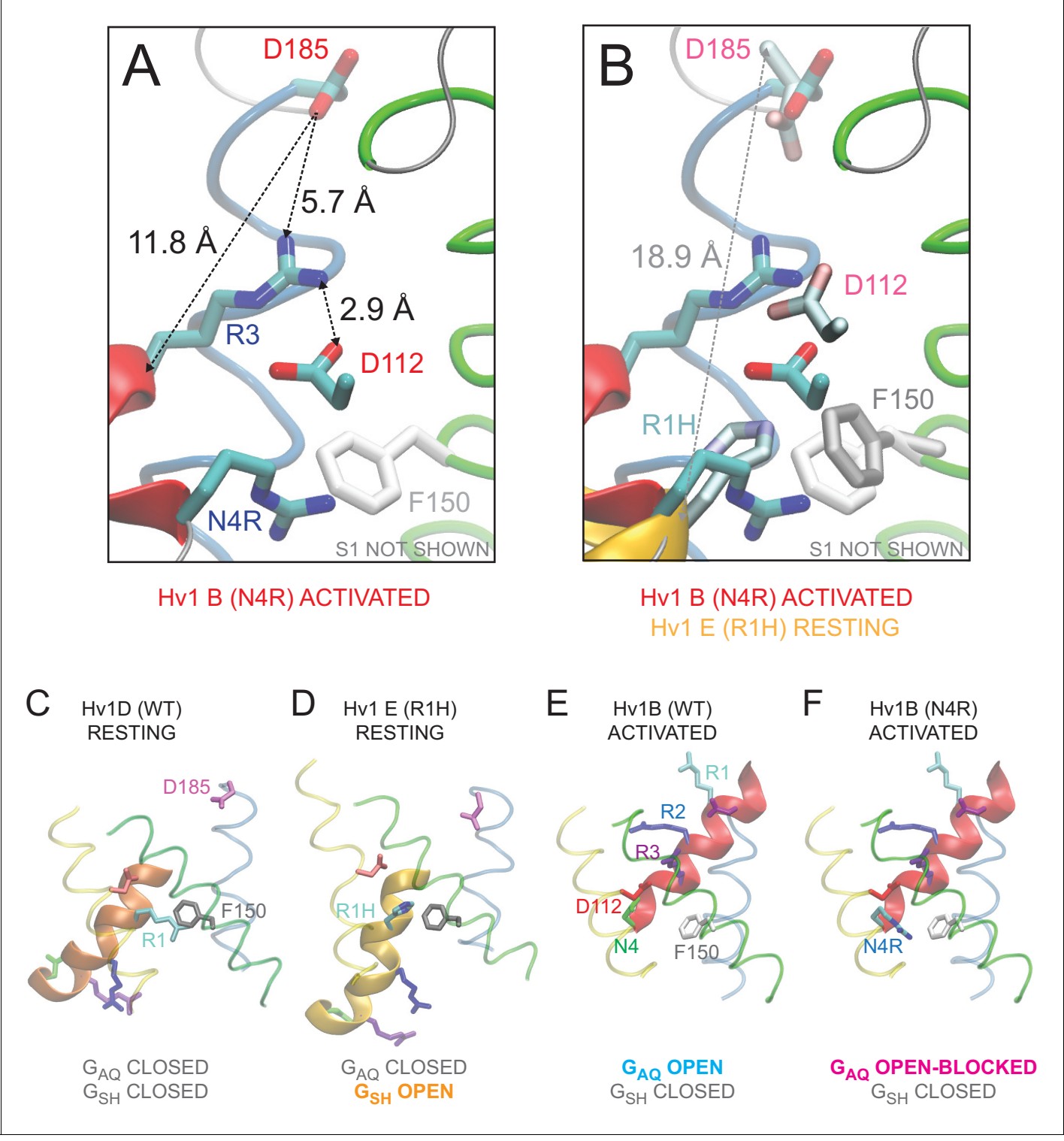

**Figure 6.** Comparison of resting- and activated-state Hv1 VS domain model structures. (**A**) Hv1 B was mutated (N4R) in silico and subjected to energy minimization to demonstrate the possible position of the N4R side chain in a VS-activated ($G_{AQ}$-blocked) conformation. Other atomic positions are not appreciably different from Hv1 B. S2 (green) and S3 (blue) helices are represented by colored tubes; S4 is shown as a red ribbon and S1 is not shown. Side chains of D112 ($D^{1.51}$), D185 ($D^{3.61}$), R3 ($R^{4.53}$) and N4R ($N^{4.56}R$) are shown in the colored licorice 'element' scheme (carbon, cyan; oxygen, red; nitrogen, blue) and the F150 ($F^{2.50}$) side chain is white. Distances (in Å) between selected carboxylate oxygen atoms in D112 or D185 and either R3 nitrogen atoms or the R3 $C_\alpha$ atom are indicated by dashed arrows. (**B**) Positions of selected residue side chains in the Hv1 E mutant model structure (produced by in silico R1H mutation of Hv1 D) are superimposed on Hv1 B N4R shown in **A**. D112, D185 and R1H side chains are represented by

*Figure 6 continued on next page*

*Figure 6 continued*

'brushed metal' coloring of licorice element representations; F150 is gray. The S4 helix in Hv1 E is shown as a gold ribbon; other helices are as shown in **A**. The dashed arrow indicates the distance (in Å) between the $C_\alpha$ atoms of D185 and R1H in Hv1 E. (**C–F**) Backbones of Hv1 D (**C**), Hv1 E (**D**), Hv1 B (**E**) and the Hv1 B N4R mutant (**F**) model structures are represented by thin (S1-S3) or thick (S4) colored ribbons and inter-helical loop regions are represented by gray tubes. Selected residue side chains are shown in colored licorice (D112/D$^{1.51}$, red; F150/F$^{2.50}$, gray or white; D185/D$^{3.61}$, magenta; R1/R$^{4.47}$, cyan; R2/R$^{4.50}$, blue; R3/R$^{4.53}$, violet; N4/N214/N$^{4.56}$, green; N4R, cyan/blue). Structures are vertically aligned by the position of the F150/F$^{2.50}$ $C_\alpha$ atom. Labels indicate the predicted functional state of the protein that correspond to the depicted structure. In **C–F**, helices are colored yellow (S1), green (S2) and blue (S3) and inter-helical loop regions are not shown for clarity; S4 residues 202–214 are colored red (Hv1 B), copper (Hv1 D) or gold (Hv1 E). *Video 4* shows Hv1 B activated- and Hv1 D resting-state model structures in rotation.

The following figure supplements are available for figure 6:

**Figure supplement 1.** Atomic distances in resting- and activated-state Hv1 VS domain model structures.

**Figure supplement 2.** Comparison of resting- and activated-state Hv1 model structures.

**Figure supplement 3.** Comparisons of putative resting- and activated-state Hv1 and *Ci* VSD model and X-ray structures.

**Figure supplement 4.** Atomic distances between backbone $C_\alpha$ atoms in resting and activated Hv1 model structures.

form a relatively immobile scaffold (*Figure 6—figure supplements 2A–E* and *3A,B*; *Video 5*). To estimate the amplitude of S4 displacement in resting- vs. activated-state VS domain X-ray and model structures, we measured distances between equivalent atoms after performing structure-based alignments. Comparing $C_\alpha$-$C_\alpha$ distances between R1 side chains in Hv1 D or Hv1 E and Hv1 B indicates that the S4 backbone is displaced ~15 Å; somewhat smaller distances (11 Å–13 Å) are measured when Hv1 D is compared to other activated-state structures (*Figure 6—figure supplements 3*, *4*). Most of the calculated difference in R1-R1 $C_\alpha$ distance is observed in the vertical (z, i.e., membrane normal) axis, but differences in helical tilt and twist are also observed (*Figure 6—figure supplements 2–4*). In summary, our comparisons of VS domain structures suggest that the S4 helix is likely to undergo an ~11–15 Å vertical translation during activation of the Hv1 VS domain.

## Discussion

The main experimental result from this study is that R205H (R1H) is sufficient to endow Hv1 with a resting-state H$^+$ shuttle conductance ($G_{SH}$). Our results are consistent with previous reports describing $G_{SH}$ in other VS domain R1H mutants (*Starace and Bezanilla, 2004*; *Struyk and Cannon, 2007*; *Villalba-Galea et al., 2013*) but contrast with a previous study in Hv1 (*Kulleperuma et al., 2013*). One possible explanation for the discrepancy is that $G_{SH}$ is difficult to measure when mutant Hv1 channel expression levels are low, as in the previous study (*Kulleperuma et al., 2013*), whereas the inducible expression system used here drives the high expression that is evidently necessary to reproducibly measure $G_{SH}$. Importantly, we show that second-site mutations (N4R and D185A or D185H) experimentally separate the $G_{SH}$-V and $G_{AQ}$-V relations, allowing us to simultaneously

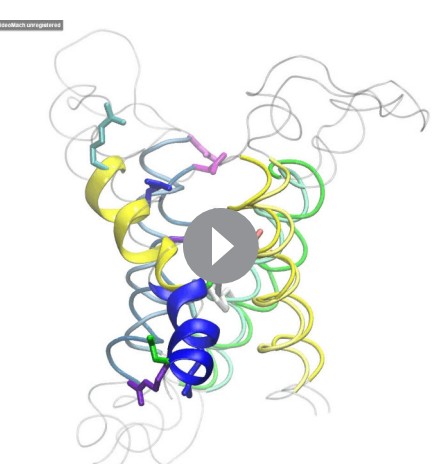

**Video 5.** Rotating view of Hv1 B activated- and Hv1 D resting-state model structures. S1-S3 helices are represented by thin colored ribbons (S1, yellow; S2, green, S3, blue), S4 segments are shown as thick colored ribbons (Hv1 B, yellow; Hv1 D, blue) and loops are shown as thin gray tubes. Selected side chains are shown in colored licorice (Hv1 B: D112, red; F150, gray; D185, magenta; R1, cyan; R2, blue; R3, violet; Hv1 D: D112, pink; F150, white; R1, cyan; R2, blue; R3, violet; N4, green). Animation shows the structures in rotation about the vertical axis.

monitor thermodynamically distinct gating transitions in Hv1. Our experimental approach may therefore be generally useful for probing structure-function relationships in VS domain-containing proteins.

Although R1H dramatically accelerates $G_{AQ}$ gating kinetics, the effect of R1H on the apparent $P_{OPEN-AQ}$-V relation is modest (*Figure 1*, *Table 2*), and $G_{AQ}$ remains $H^+$-selective (*Figure 1—figure supplement 2*) (*Kulleperuma et al., 2013*). $G_{SH}$ in Shaker R1H is also $H^+$-selective (*Starace and Bezanilla, 2004*), and the sensitivity of inward resting-state current amplitude to changes in $pH_O$ indicates that $G_{SH}$ in Hv1 (*Figure 4*), *Ci* VSP (*Villalba-Galea et al., 2013*) and Shaker (*Starace and Bezanilla, 2004*) R1H mutants are likely to utilize a shared mechanism. A simple explanation for the available data is that R1H mutations primarily affect side chain $pK_a$, allowing channel-like proton shuttling in the resting state without substantially affecting protein structure or VS activation mechanism. R1 mutation to other side chains (i.e., R1A/C/Q/S) confers a resting-state 'omega' conductance ($G_\Omega$) that is permeable to small monovalent cations ($Na^+$ and $K^+$) and thus distinct from $G_{SH}$ (*Gosselin-Badaroudine et al., 2012*; *Tombola et al., 2005*; *Capes et al., 2012*; *Gamal El-Din et al., 2010*, *2014*; *Sokolov et al., 2005*). For reasons that remain unclear, R1A and R1Q are insufficient to confer $G_\Omega$ in Hv1 (*Ramsey et al., 2006*; *Sasaki et al., 2006*). Further studies are also needed to determine whether Hv1 R2H or R3H mutant proteins mediate carrier-like ($G_{CA}$) conducting states similar to those reported in Shaker (*Starace and Bezanilla, 2001*; *Starace et al., 1997*). We conclude that $G_{AQ}$, $G_{SH}$, $G_\Omega$ and $G_{CA}$ reflect distinct types of 'gating pore' conductances ($G_{GP}$), and that each exhibits characteristic gating and ion permeation properties which can be experimentally exploited to interrogate resting-state structure-function relationships.

## Resting-state $H^+$ shuttling in Hv1 R1H

Among various $G_{GP}$, $G_{SH}$ measurement has unique properties that offer deep insight into VS activation mechanism and structure: (**1**) The sufficiency of R1H to confer $G_{SH}$ implies that the introduced His imidazole side chain 'short-circuits' a highly focused electrical field in the VS domain resting conformation (*Starace and Bezanilla, 2004*; *Villalba-Galea et al., 2013*; *Starace et al., 1997*). (**2**) R1 appears to contribute ~1 $e_0$ to the gating valence in both Shaker and Hv1 (*Gonzalez et al., 2013*; *Seoh et al., 1996*; *Aggarwal and MacKinnon, 1996*), and $G_{SH}$ gating exhibits a similarly small (~0.7 $e_0$) apparent valence, constraining possible side chain positions within the electric field (*Gonzalez et al., 2013*; *Ahern and Horn, 2005*; *Tao et al., 2010*). (**3**) Voltage-dependent block of $G_{AQ}$ by N4R places terminal nitrogen atoms at the intracellular entrance of the $H^+$ permeation pathway and therefore close to the hydrophobic barrier formed by conserved hydrophobic side chains, including $F^{2.50}$. Biophysical properties of R1H and the effects of second-site mutations can thus be used to experimentally constrain the relative positions of specific side chains in VS domain model structures (*Figure 6—figure supplements 2–4*).

Our comparison of new and existing resting-state VS domain model and X-ray structures highlights structural features that are required for $G_{SH}$. The VS domain contains an hourglass-shaped aqueous central crevice with a central hydrophobic barrier (*Ramsey et al., 2010*; *Takeshita et al., 2014*; *Wood et al., 2012*; *Chamberlin et al., 2014*; *Kulleperuma et al., 2013*). The electrical field is highly focused across the hydrophobic barrier, and side chain chemistry at this location is therefore exquisitely sensitive to changes in membrane potential (*Lacroix et al., 2014*; *Vargas et al., 2012*; *Tao et al., 2010*). Although VS domains share a common protein fold, subtle differences in local structure and chemistry have the potential to imbue different voltage sensors with divergent functional properties (i.e., $H^+$ permeation or pH-dependent gating). A detailed understanding of the similarities and differences in VS domain structure is therefore essential for dissecting VS mechanism.

## Hv1 VS domain resting-state structure

Grotthuss-type $H^+$ shuttling by the R1H imidazole side chain demonstrates that in the resting-state conformation, R1 is located at the hydrophobic constriction and the central crevice is hydrated and accessible to both intra- and extra-cellular water molecules. Resting-state VS domain structures (*Figures 5*, *6*) in which the R1 side chain extends away from $F^{2.50}$ and into the extracellular vestibule (*Takeshita et al., 2014*; *Li et al., 2014*; *Jensen et al., 2012*; *Delemotte et al., 2011*; *Li et al., 2015*) may therefore represent intermediate-state conformations rather than the full resting-state conformation. A distinguishing feature of the Hv1 D resting state model and *At* TPC1 DII VS domain

X-ray structures is the orientation of the R1 side chain, which extends into the intracellular vestibule (*Figure 5*). The position of the R1 side chain is consistent with the hypothesis that a local, voltage-dependent conformational rearrangement of R1 (or R1H) constitutes an initial step in the VS activation pathway, and that $G_{SH}$ gating directly reports this transition.

Consistent with our data, D233/D$^{3.61}$ and R1/R255/R$^{4.47}$ are distant in the *Ci* Hv1 resting-state model (*Chamberlin et al., 2014*); however, the R255 side chain is intracellular to F$^{2.50}$ in the *Ci* Hv1 model and R1 does not appear to be appropriately positioned to mediate $G_{SH}$ if it were mutated to His (the ability of R1H mutation to confer $G_{SH}$ in *Ci* Hv1 remains to be tested experimentally). In mHv1cc, the D181/D$^{1.61}$ faces away from both R1/R201/R$^{4.47}$ and R2/R204/R$^{4.50}$, and these ionizable side chains are uncharacteristically packed into hydrophobic crevices (*Figures 6A*; *Figure 5—figure supplement 4A*; *Video 2*) (*Takeshita et al., 2014*). Finding these ionizable side chains in hydrophobic environments is unexpected because R1-R3 are expected to contribute cationic gating charge (*Gonzalez et al., 2013*; *Seoh et al., 1996*; *Aggarwal and MacKinnon, 1996*) and D$^{1.51}$ and D$^{3.61}$ appear to engage in Coulombic interactions that stabilize the activated-state conformation of the Hv1 VS domain (see below). In contrast to mHv1cc, D/N$^{1.51}$, D/E$^{3.61}$, R$^{4.47}$, R$^{4.50}$ and R$^{4.43}$ are readily solvent-accessible in Hv1 D and *At* TPC1 DII VS domains.

## Hv1 VS domain activated-state structure

Outward current carried by $G_{AQ}$ is selectively blocked in R1H-N4R, and rapid (< 1 ms) relief of block upon subsequent hyperpolarization (*Figure 2A*) strongly implicates that the N4R side chain functions as a tethered blocker operating from the intracellular side of the H$^+$ permeation pathway. In agreement with a widely-accepted prevailing model of structural rearrangement during VS activation, we find that the position of F$^{2.50}$ is similar in $G_{AQ}$-open (Hv1 B) and $G_{SH}$-open (Hv1 D) model structures, and the central hydrophobic barrier is evidently maintained throughout the Hv1 gating cycle. Nonetheless, the gating pore remains well-hydrated in both resting and activated-state conformations (*Figure 5—figure supplement 3*). By analogy, we hypothesize that R3 needs to move outward, past F$^{2.50}$, to unblock the central crevice and open $G_{AQ}$. The dramatic positive shifts in $G_{AQ}$-V relations imparted by D185/D$^{3.61}$ (*Figure 3*) and R3 (*Ramsey et al., 2010*) mutations argue that interactions between these side chain are required for activated-state stabilization in WT Hv1 channels.

Consistent with the observation that D112/D$^{1.51}$ mutations also cause large positive shifts in the $G_{AQ}$-V relation (*Ramsey et al., 2010*), we find that R3 also interacts with D112 in the Hv1 B model structure. D185 mutations do not alter $G_{SH}$-V gating, indicating that this residue does not meaningfully contribute to stabilization of the Hv1 VS resting-state conformation, and D185 is appropriately distant from R1 in the Hv1 D resting-state model. In contrast to our experimental observations, a D185-R1 interaction is predicted to stabilize the Hv1 FL model activated-state conformation (*Li et al., 2015*). We conclude that $G_{AQ}$ opening is directly controlled by a late step in the Hv1 activation pathway that requires interactions between D185 and one or more S4 Arg residues, most likely R3. We hypothesize that D185/$^{D3.61}$ functions to pull S4 upward, and thus helps to stabilize the $G_{AQ}$-open conformation. D112/D$^{1.51}$ also appears to play an important role in stabilizing S4 in the $G_{AQ}$-permissive conformation of S4, and may indirectly interact with D185 through R3, as seen in Hv1 B N4R (*Figure 6—figure supplement 1*). D$^{1.51}$ and D$^{3.61}$ are selectively conserved in Hv1 channel VS domains (*Figure 1—figure supplement 1*), and their contributions to activated-state stabilization are predicted to be necessary for H$^+$ channel activity.

$G_{AQ}$ closing requires only a small inward translation of R3 toward F$^{2.50}$. The R3-associated cation may disrupt the hydrogen bond network required for H$^+$ transfer and/or electrostatically prevent inward H$^+$ flux through the central crevice. Membrane hyperpolarization presumably drives the VS through several non-conducting intermediate states similar to those seen in Kv channel VS domain simulations (*Jensen et al., 2012*; *Delemotte et al., 2011*), and the full resting conformation is achieved when R1 reaches the position near F$^{2.50}$ seen in Hv1 D (*Figure 5E*). The mechanism outlined above is generally consistent with a widely accepted model of the VS activation process (*Vargas et al., 2012*), and $G_{SH}$ data reported here extend this model to Hv1 channel gating.

An intriguing but as yet unresolved question is whether the amplitude of S4 movement is similar in Hv1, Ci VSP and voltage-gated channels like Shaker and Kv1.2. The gating valence in Shaker K$^+$ channels is ~3 electronic charges (e$_0$) per VS domain, and likely reflects the movement of R1-R4 side chains in or through the electrical field (*Bezanilla, 2008*; *Seoh et al., 1996*; *Aggarwal and*

*MacKinnon, 1996*). A limiting slope analysis of Hv1 gating suggests that the effective gating valence (~2.5 $e_0$/VS) is slightly smaller than Shaker (*Gonzalez et al., 2013*), consistent with the substitution of a neutral polar Asn ($N^{4.56}$; N214 or N4 in Hv1) at the R4 position (*Figure 1A*, *Figure 1—figure supplement 1*). The decreased gating charge in Hv1 suggests that VS activation (and thus $G_{AQ}$ opening) might require a smaller displacement of S4 than is seen in prototypical VGCs like Shaker. However, except for state-dependent mapping of chemical sensitivity in *Ci* Hv1 (*Gonzalez et al., 2013*), experimental data that constrain S4 position in resting- and activated-state conformations of the Hv1 VS domain have not been reported.

Atomic distances measured in resting- and activated-state Hv1 models suggest that R1 ($R^{4.47}$) $C_\alpha$ atoms in S4 could move as much as 14–16 Å (*Figure 6—figure supplements 3* and *4*). Shorter distances (11–13 Å) are measured when Hv1 D is compared to other activated-state VS domain models (*Figure 6—figure supplements 3* and *4*). The apparent flexibility of Arg side chains in VS domains (*Li et al., 2014*) suggests that the magnitude of S4 translation may not be easily inferred from measurements of gating valence alone. Proton transfer via $G_{SH}$ and voltage-dependent block of $G_{AQ}$ appear to place stringent constraints on the relative positions of target side chain atoms, and may offer advantages over alternative approaches, such as chemical accessibility in Cys mutant proteins, for ascertaining structural changes that occur during VS activation. However, a systematic comparison of experimental and structural strategies in each model system is needed to identify specific advantages and liabilities of various approaches. The combination of electrophysiological and computational approaches used here allows researchers to iteratively refine model structures and experimentally interrogate new structure-based hypotheses of mechanism in the context of biophysically-determined kinetic and thermodynamic parameters of protein function, and is thus faster and more flexible than structural determination by X-ray crystallography alone. Although our experimental data probably do not offer sufficient spatial resolution to discriminate whether S4 moves ~12 Å vs. ~14 Å, it is difficult to reconcile our data with models in which S4 movement is closer to 5 Å, such as is seen when *Ci* VSD$_U$ and *Ci* VSD$_D$ X-ray structures are compared (*Figure 6—figure supplements 3* and *4*) (*Li et al., 2014*).

Direct comparisons of $G_{AQ}$ and $G_{SH}$ gating reveal additional insight into the VS activation mechanism. $G_{AQ}$-V and $G_{SH}$-V relations are oppositely sensitive to changes in membrane potential and gated over widely-separated ranges of voltage change, and thus report thermodynamically distinct gating transitions. We show for the first time that $G_{AQ}$ and $G_{SH}$ gating is similarly sensitive to changes in pH$_O$ (*Figure 4*). In a previously proposed Hv1 gating scheme, the pH dependence of $G_{AQ}$ gating attributed to closed-state transitions that occur early in the Hv1 activation pathway (*Villalba-Galea, 2014*), and the pH$_O$ dependence of $G_{SH}$ gating reported here is consistent with this model. Voltage clamp fluorimetry (VCF) in *Ci* Hv1 also supports the conclusion that VS conformational rearrangements are detectable prior to $G_{AQ}$ opening (*Qiu et al., 2013*), but the pH dependence of fluorescence changes has not been investigated in Hv1. Intriguingly, a VCF study conducted in hERG (*Shi et al., 2014*) suggests that pH-dependent gating could be a more widespread property of VS activation mechanism than has previously been appreciated.

The mechanism of pH$_O$-dependent gating in Hv1 is enigmatic. pH$_O$ sensitivity is surprisingly refractory to neutralizing mutagenesis of ionizable residues in Hv1 (*Ramsey et al., 2010*; *Musset et al., 2011*; *Morgan et al., 2013*). Recently, W207/$W^{4.49}$ mutations were shown to alter the pH sensitivity of $G_{AQ}$ gating at alkaline pH$_O$, but pH-dependent gating at physiological pH$_O$ is similar to WT Hv1 (*Cherny et al., 2015*). W207 is not predicted to face the hydrated central crevice in either resting- or activated-state Hv1 VS domain models, and the mechanism by which W207X mutations affect pH-dependent $G_{AQ}$ gating remains mysterious (*Cherny et al., 2015*). Given that $G_{SH}$ and $G_{AQ}$ appear to share the requirement for a hydrated central crevice $H^+$ permeation, a plausible hypothesis is that changes in pH$_O$ *or* pH$_I$ exert their effects mainly by affecting hydrogen bonding patterns in the central crevice. For example, pH-dependent changes in Coulombic interactions within the extracellular vestibule could be coupled with reciprocal conformational changes in the structure of the intracellular electrostatic network, thus altering the VS resting-activated equilibrium. However, the mechanism of pH-dependent conformational coupling remains to be elaborated.

## Proton conduction and selectivity in Hv1

The difference in the apparent maximal amplitudes of $G_{SH}$ and $G_{AQ}$ suggests that the mechanisms of $H^+$ transfer could be distinct. We and others previously hypothesized that proton permeation via

$G_{AQ}$ occurs in a water wire (*Ramsey et al., 2010*; *Wood et al., 2012*; *Freites et al., 2006*); DeCoursey and colleagues subsequently argued side chain ionization of D112/D$^{1.51}$ is required for H$^+$ transfer (*Musset et al., 2011*; *Dudev et al., 2015*). If proton transfer via $G_{AQ}$ and $G_{SH}$ operate by a 'shuttle' mechanism requiring explicit ionization of D112 or R1H, respectively, we might expect the unitary conductances ($\gamma_{AQ}$ and $\gamma_{SH}$) to be similar. If $G_{AQ} = N \cdot \gamma_{AQ} P_{OPEN-AQ}$ and $G_{SH} = N \cdot \gamma_{SH} \cdot P_{OPEN-SH}$, the observation that apparent $G_{AQmax}$ is ~five-fold larger than $G_{SHmax}$ (*Figure 2*) argues that $\gamma_{AQ} \approx 5 \cdot \gamma_{SH}$. The smaller $G_{SH}$ unitary H$^+$ transfer capacity is consistent with the hypothesis that hydrogen bonds, which are necessary for H$^+$shuttling, are constrained by H$^+$ donor and acceptor atom geometry and distance (*Cherny et al., 2015*). Although D112/D$^{1.51}$ is necessary for maintaining the exquisitely high H$^+$ selectivity measured in WT Hv1 (*Dudev et al., 2015*), necessity for D112/D$^{1.51}$ to directly catalyze $G_{AQ}$ H$^+$ transfer (*Dudev et al., 2015*) has not been experimentally determined, and a water-wire mechanism for $G_{AQ}$ (*Ramsey et al., 2010*) is equally compatible with the available experimental data. We hypothesize that $G_{AQ}$ utilizes ensemble of highly dynamic hydrogen bonds between and among waters and protein atoms diffusive in the central crevice for H$^+$ transfer in a water wire. Functional redundancy imbued by a water wire is consistent with the resiliency of Hv1 to mutagenesis and potentially explains the more robust H$^+$ transfer capacity of $G_{AQ}$.

Our experimental and computational results suggest a mechanism for H$^+$ conduction and selectivity in Hv1 that is distinct from the interpretation of Dudev, et al. (*Dudev et al., 2015*). An acidic residue in S1 (D$^{1.51}$; E$^{1.51}$ or D$^{1.55}$ in mutant channels) located in the hydrated central crevice prevents permeation of solution anions (i.e., Cl$^-$, MeSO$_3^-$ or OH$^-$) while R3/R$^{4.53}$ limits cation (Li$^+$ or Na$^+$) permeability (*Musset et al., 2011*; *Berger and Isacoff, 2011*; *Morgan et al., 2013*). D112/D$^{1.51}$ and R3/R$^{4.53}$ mutations allow ions other than H$^+$ to permeate, as reported (*Musset et al., 2011*; *Berger and Isacoff, 2011*), demonstrating that these side chains remain ionized in WT Hv1 when $G_{AQ}$ is open. Because monovalent ions (other than H$^+$) are unlikely to permeate as dehydrated ions in D112/D$^{1.51}$ and R3/R$^{4.53}$ mutant channels, the central crevice remains well-hydrated in these mutant channels (*Ramsey et al., 2010*). We hypothesize that the previously proposed water-wire mechanism for $G_{AQ}$ remains operational in D112/D$^{1.51}$ and R3/R$^{4.53}$ mutants, but permeating ions like Na$^+$ and Cl$^-$ transiently disrupt the hydrogen bond structure that is necessary for Grotthuss-type H$^+$ transfer via $G_{AQ}$. However, intervals between diffusive ion permeation events, rapid H$^+$ transfer in the water wire continues unabated. The eroded selectivity reported for D112 and R3 mutants therefore reflects the time-averaged amalgam of two distinct conduction mechanisms: 1) monovalent ion diffusion through a water-filled gating pore, and 2) Grotthuss-type proton transfer in a water wire. In short, $G_{AQ}$ in both WT and mutant Hv1 channels is mediated water-wire proton transfer, but mutant channels allow more diffusive anion/cation leakage through the hydrated central crevice.

Taken together, our experimental data and model structures indicate that $G_{AQ}$ and $G_{SH}$ share a common H$^+$ permeation pathway within the hydrated central crevice, but the underlying mechanisms of H$^+$ transfer are distinct. A water wire supports $G_{AQ}$, while H$^+$ shuttling via $G_{SH}$ requires explicit ionization of the introduced His side chain. The unitary conductance of $G_{SH}$, which is ~5 times smaller than $G_{AQ}$, reflects the additional complexity that is inherent to the H$^+$ shuttle process. The His side chain must first accept a proton from water in the extracellular vestibule, likely undergo a rotation or tautomerization event that delivers the associated proton across the hydrophobic barrier, donate H$^+$ to water in the intracellular vestibule, and finally return to the initial conformation to repeat the cycle. The H$^+$ shuttle mechanism is channel-like in the sense that voltage-dependent conformational changes gate $G_{SH}$ and the $I_{STEP}$-V relation appears linear (Ohmic) at large negative voltages but transporter-like with respect to the necessity for side chain ionization. $G_{AQ}$, on the other hand, requires only water molecules, and the myriad possible hydrogen bonding patterns within the hydrated crevice confers a functionally robust, rapid, and H$^+$-selective proton transfer pathway. Systematic testing of the hypotheses elaborated here will require additional computational and experimental strategies, but the results of future studies are likely to produce fundamentally important insights into the mechanisms of VS activation gating by changes in voltage and pH gradients and strategies that underlie H$^+$-selective transport in VS domains and other protein systems.

## Materials and methods

### Molecular biology and cell lines

Human Hv1 cDNA (NM_032369) carrying an N-terminal Venus tag was subcloned from pBSTA (gift of Carlos A. Villalba-Galea) into pcDNA5/FRT/TO using standard methods and isogenic tetracycline-inducible FlpIn293-TREx stable cell lines were generated according to the manufacturer's directions (ThermoFisher Scientific, Waltham, MA ). Parental FlpIn293-TREx cells were obtained directly from the manufacturer and cultured as instructed; cells were not independently authenticated or tested for mycoplasma. Hygromycin B (100 μg/ml) was used for selection and propagation of isogenic stable cell lines. Cells were plated onto glass coverslips and expression of mutant Hv1 proteins was induced by addition of tetracycline (0.5–1 μg/ml) to the culture medium 12–48 hr prior to electrophysiology. Close to 100% of tetracycline-induced cells typically express Venus fluorescence, and both the intensity and pattern of fluorescence was similar among cells expressing a given mutation. Absolute current amplitudes appeared to correlate positively with increasing [tetracycline] and induction time, although this pattern was not studied systematically.

### Electrophysiology

Whole-cell currents were measured at 22–24°C using a List EPC-7 or A-M Systems model 2400 amplifier. Data were low-pass filtered at 2–5 kHz digitized at 10–20 kHz using a National Instruments USB-5221 or USB-5251 DAQ interfaced to a PC computer running a custom LabVIEW 7-based data acquisition and amplifier control program (C. A. Villalba-Galea; details and software distribution available on request). Data were analyzed using Clampfit9 (Molecular Devices) and Origin 6.0 (Microcal). The standard intracellular and extracellular solutions contained (in mM): 100 Bis (2-hydroxyethyl) amino-tris(hydroxymethyl) methane (Bis-Tris), 1 ethylene glycol tetraacetic acid (EGTA), 8 HCl and pH was adjusted to 6.5 and final osmolality of 310–320 mOsm by addition of tetramethylammonium hydroxide (TMA·OH) and methanesulfonic acid (HMeSO$_3$). Current reversal potentials and pH$_O$-dependent gating were measured in bath solutions containing either 100 mM 2-(N-morpholino) ethanesulfonic acid (MES, pH 5.5) or 4-(2-hydroxyethyl)-1-piperazineethanesulfonic acid (HEPES), pH 7.5 in place of Bis-Tris, as previously described (*Ramsey et al., 2010*). Series resistance was not routinely compensated and liquid junction potential corrections are not applied.

### Data analysis

Unless otherwise indicated, the data represent means ± SEM of values measured in $n$ cells. I$_{STEP}$ represents the peak current during steps to the indicated potentials. In most cases, I$_{STEP}$ was stable, but in cells with large currents we sometimes observed a decay in the amplitude of I$_{STEP}$ during the voltage step that we attribute to a change in the pH gradient, which may not be sufficiently controlled by 100 mM pH buffer in the recording solutions when G$_{SH}$ is open. I$_{STEP}$ is measured during voltage steps and I$_{TAIL}$ represents peak current immediately after a subsequent hyperpolarizing step determined by fitting current time course to a single exponential function G$_{STEP}$ was calculated from G$_{STEP}$ = I$_{STEP}$/V-E$_{REV}$ where E$_{REV}$ is the zero-current potential determined from inspection of the I$_{STEP}$-V relation. I$_{TAIL}$ amplitude is determined by fitting current decay to a mono-exponential function of the form I$_{TAIL}$ = I$_0$ + Ae$^{-V/t}$ (where I$_0$ is the minimum current after decay of I$_{TAIL}$, A is current amplitude, V is membrane potential and $t$ is time) and extrapolating fits to the instant at which the voltage was changed. V$_{THR}$, the apparent threshold for activation of I$_{TAIL}$ is estimated from visual inspection of I$_{TAIL}$ as previously described (*Musset et al., 2008*). Steady-state conductance during voltage steps (G$_{STEP}$) is calculated from G$_{STEP}$ = I$_{STEP}$/V-E$_{REV}$ where E$_{REV}$ is the zero-current potential determined from inspection of the I$_{STEP}$-V relation. In some experiments (see *Figure 1—figure supplement 2*), we changed V$_{TAIL}$ (following a constant V$_{STEP}$) to determine E$_{REV}$ of tail currents as previously described (*Ramsey et al., 2006*). Offline linear leak subtraction of I$_{STEP}$-V relations was performed only in cases where the I-V relations are clearly linear (i.e., I$_{STEP}$-V in R1H-N4R at V$_m$ > 0 mV and I$_{TAIL}$-V in R1H or R1H-N4R at V$_m$ < −50 mV). I$_{TAIL}$-V relations are fit to a Boltzmann function of the form: $I_{TAIL} = \frac{(I_{TAIL\max}) - (I_{TAIL\min})}{1 + e^{V - V_{0.5}dx}} + I_{TAIL\min}$, where V$_{0.5}$ is the voltage at which 50% of the maximum current is reached, dx is a slope factor, and I$_{TAILmax}$ and I$_{TAILmin}$ represent the maximum and minimum tail current amplitudes, respectively. G$_{STEP}$-V relations are fit to a single Boltzmann of the form $G_{STEP} = \frac{(G_{STEP\max}) - (G_{STEP\min})}{1 + e^{V - V_{0.5}dx}} + G_{STEP\min}$ where V$_{0.5}$, dx, G$_{STEPmax}$ and G$_{STEPmin}$ have the same meanings as

defined for $I_{TAIL}$. In some cases, effective gating valence ($z_G$) was calculated from fitted $dx$ values by $z_G = RT/F \cdot dx$, where F, R and T have their usual meanings (e.g., RT/F = 25.3 mV at 20°C). $dG_{STEP}/dV$ relations are fit to a Gaussian function of the form $dG_{STEP}/dV = (dG_{STEP}/dV)_0 + \left(\frac{A}{\varpi\sqrt{\pi/2}}\right)e^{-2\frac{(V-V_{PEAK})^2}{\varpi^2}}$, where $V_{PEAK}$ is the voltage at which the function reaches its maximum and ω is a width factor.

## Homology modeling and simulation

Models for Hv1 in putative activated (Hv1 C) and resting (Hv1 D) states were developed from the Kv1.2 X-ray structure (pdb:3LUT) and resting state model structure of the Shaker voltage-gated K+ channel (*Pathak et al., 2007*) templates, respectively, using standard homology modelling procedures as described previously (*Ramsey et al., 2010*; *Mokrab and Sansom, 2011*). Hv1 B model construction was published previously (*Ramsey et al., 2010*). Briefly, homologous sequences were obtained for the target sequences and structures from UniRef100 (*Bairoch et al., 2005*) using non-iterative BLAST (e-value < 10). The two proteins were aligned using MAFFT (*Katoh et al., 2002*) based on the BLOSUM62 substitution matrix (*Henikoff and Henikoff, 1992*). Next, a structural profile (i.e. Position-Specific Substitution Matrices - PSSMs) was calculated for the structure and a sequence profile for the target sequence was created. The structural profile was then aligned against the sequence profile using FUGUE (*Shi et al., 2001*). The resulting structure-sequence alignment was manually adjusted to ensure conservation of key residues, then used as input for MODELLER (*Sali and Blundell, 1993*) to generate ten models per alignment. The best models were selected based on the energy and constraint violation values of MODELLER and the sequence-structure compatibility scores of pG (*Sánchez and Sali, 1998*), PROSA2003 (https://prosa.services.came.sbg.ac.at/prosa.php) (*Sippl, 1993*) and VERIFY3D (http://nihserver.mbi.ucla.edu/Verify_3D/) (*Lüthy et al., 1992*) as previously described (*Ramsey et al., 2010*; *Mokrab and Sansom, 2011*). Any unreliable regions in the model were improved by altering the alignments manually using ViTO (http://abcis.cbs.cnrs.fr/VITO/DOC/index.html) (*Catherinot and Labesse, 2004*).

All-atom molecular dynamics (MD) simulations were prepared as described (*Sands and Sansom, 2007*). Side-chain ionization states were determined based on $pK_a$ calculations performed using PROPKA (http://propka.ki.ku.dk/). Ionizable residues were predicted to be in the default states at pH 7 based on standard $pK_a$ values for each residue. We adopted lipid parameters as used previously (*Berger et al., 1997*). Prior to the production run, a 1 ns equilibration run was performed during which all of the heavy (i.e., not H+) protein atoms were harmonically restrained with an isotropic force constant of 1000 kJ mol$^{-1}$ nm$^{-1}$. Restrained MD runs were performed at 300K for each protein-bilayer system. Finally, all positional restraints were removed and 20 ns duration production run simulations were performed for each system. MD simulations were performed using GROMACS 3.3 (*Van Der Spoel et al., 2005*), implementing the GROMOS96 force field (http://www.gromos.net). Lipid parameters were based on GROMOS96, supplemented with additional bond, angle and dihedral terms (*Berger et al., 1997*). All energy minimization procedures used < 1000 steps of the steepest descent method in order to relax any steric conflicts generated during system setup. Long-range electrostatic interactions were calculated using the particle mesh Ewald (PME) method, with a 12 Å cutoff for the real space calculation (*Sagui et al., 2004*). A cut-off of 12 Å was used for the van der Waals interactions. The simulations were performed at constant temperature, pressure and number of particles. The temperature of the protein, lipid and solvent (waters and ions) were separately coupled using the Nosé-Hoover thermostat (*Popov and Knyazev, 2014*) at 310°K, with a coupling constant, $\tau_T$ = 0.1 ps. System pressures were semi-isotropically coupled using the Parrinello-Rahman barostat (*Parrinello and Rahman, 1981*) at 1 bar with a coupling constant, $\tau_P$ = 1 ps and compressibility = 4.5 × 10$^{-5}$ bar$^{-1}$. The LINCS algorithm (*Hess, 2008*) was used throughout to constrain bond lengths. The time step for integration in both simulations was 2 fs. All analyses used GROMACS tools and locally written code.

The final snapshots from Hv1 D GROMOS96 MD simulation was used as the template for the introduction of the R1H mutation using the Mutator plugin (VMD 1.9.2). The final snapshot of the GROMOS96 MD of the Hv1 B model structure (*Ramsey et al., 2010*) was used as the template for production of Hv1 B N4R using Modeller 9.16 (*Sali and Blundell, 1993*). All side chains were assumed to have the typical solution $pK_a$ defined in PROPKA (*Dolinsky et al., 2004*; *Olsson et al., 2011*), and His residues were modeled with the delta nitrogen (HSD) protonated, which was the ionization state predicted by PROPKA. WT and mutant resting- and activated-state models were

subsequently imbedded in a POPC membrane and solvated with a 150 mM KCl solution and energy-minimized in order to remove any unfavorable contacts. After energy minimization POPC lipid tails were allowed to equilibrate around the protein for 0.5 ns, after which the system was simulated according the NPT ensemble with harmonic constraints (5 kcal/mol·Å) applied to the alpha carbons for 1.5 ns. Once the system reached equilibrium, as judged by protein RMSD, stable system volume, and converged energy terms, Hv1-POPC systems were then simulated for 10 ns with the NPT ensemble at 300K. All energy minimizations were carried out in 5000 steps using conjugate gradient and line search algorithms. Simulations were carried out according to the CHARMM36 force field (*Best et al., 2012*) with the NPT ensemble at 300K and 1 bar using a CUDA build of NAMD 2.10 (*Phillips et al., 2005*) on a GPU server. Long range electrostatic interactions were calculated using a PME method with a 12 Å cutoff distance. Constant temperature is accomplished using Langevin dynamics and constant pressure control is accomplished using a Nose-Hoover Langevin piston. 2 fs time steps were used. All analysis was carried out in VMD1.9.2. Protein structures were aligned using MultiSeq STAMP (*Roberts et al., 2006*) implemented in VMD1.9.2; structures in *Figure 5—figure supplement 2* were aligned using DeepAlign (http://raptorx.uchicago.edu/DeepAlign/submit/). Structural comparisons to Hv1 FL were conducted on chain A of the dimer, which is not identical to chain B (*Li et al., 2015*). Coordinates for *Ci* VSD$_U$ (pdb: 4G7V), *Ci* VSD$_D$ (pdb: 4G80), mHv1cc chimera (pdb: 3WKV), Kv1.2 (pdb: 3LUT) and *At* TPC1 DII (pdb: 5E1J and 5DQQ) VS domain X-ray structures are available at http://www.rcsb.org.

## Acknowledgements

The authors wish to thank Wendy Calchary, Audrey Le and Xuanyu Meng for technical assistance and Louis J De Felice and Carlos A Villalba-Galea for helpful discussions.

## Additional information

### Funding

| Funder | Grant reference number | Author |
| --- | --- | --- |
| National Institute of General Medical Sciences | R01GM092908 | Aaron L Randolph Ashley L Bennett Ian Scott Ramsey |
| Wellcome Trust | | Younes Mokrab Mark SP Sansom |

The funders had no role in study design, data collection and interpretation, or the decision to submit the work for publication.

### Author contributions

ALR, Designed and analyzed experiments, Conception and design, Acquisition of data, Analysis and interpretation of data, Drafting or revising the article, Contributed unpublished essential data or reagents; YM, ALB, MSPS, Conceived, designed, analyzed model structures, Conception and design, Acquisition of data, Analysis and interpretation of data, Drafting or revising the article, Contributed unpublished essential data or reagents; ISR, Conceived, designed, analyzed and interpreted experiments, Analyzed model structures and prepared graphical renderings, Conception and design, Acquisition of data, Analysis and interpretation of data, Drafting or revising the article, Contributed unpublished essential data or reagents

### Author ORCIDs

Younes Mokrab, http://orcid.org/0000-0003-1611-6692
Mark SP Sansom, http://orcid.org/0000-0001-6360-7959
Ian Scott Ramsey, http://orcid.org/0000-0002-6432-4253

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
