## [Decision Letter]

Thank you for submitting your article "Proton currents constrain structural models of voltage sensor activation" for consideration by *eLife*. Your article has been favorably evaluated by Gary Westbrook (Senior editor) and three reviewers, one of whom, Kenton J Swartz (Reviewer #1), is a member of our Board of Reviewing Editors. The following individual involved in review of your submission has agreed to reveal their identity: David E Clapham (Reviewer #3). The reviewers have discussed the reviews with one another and the Reviewing Editor has drafted this decision to help you prepare a revised submission.

Summary:

This is an interesting manuscript describing the creation and careful characterization of a proton shuttle in the resting state of the voltage-activated proton channel (Hv1) by introduction of a His residue at the outer R1 position within the S4 helix. The authors have done all the right experiments to understand the R1H mutant, the experimental data appear to have been very thoughtfully collected and analyzed, and are of very high quality. The significance of the proton shuttle is that it positions the R1 position of S4 down near the charge transfer Phe in S2 where one could readily imagine protons hopping between external and internal solutions. The authors also create new models for the resting and activated states of Hv1, and these are compared with structures and models of Hv1 and other proteins containing S1-S4 domains. Overall we think the work is appropriate for *eLife*, but the text and computational components will require careful revision.

Essential revisions:

1) Throughout the manuscript the authors complicate the story by assuming that the voltage-dependence for G_SH_ must reflect a conformational change in the protein, rather than arising from an intrinsic rectification to proton permeation. Until we get to Figure 4, all the data actually would support a mechanism involving rectification of permeation. Also, the apparent shifts in the dG/dV analysis are only a hint. We suggest removing all verbiage about gating mechanisms surrounding G_SH_ until you get to the section of the Results where you wish to present the Figure 4 experiment. Then you can raise the issue about mechanism of voltage-dependence to G_SH_, remind the reader about mutants affecting G_AQ_ and not G_SH_, and then show how pH seems to shift both and talk about what that might mean. The authors ideas may be on the right track, but they don't have very firm ground to stand on and the presentation should reflect that.

2) The paper is straightforward to read and understand for experts, but it is too longwinded and will be impenetrable for the outsider. Terms like Ftail and Fhook are thrown about and won't be clear to those who haven't read the Larsson paper. In its present form it will probably only be read by scientists working on Hv1 or those who are total nerds and simply love the topic of voltage sensing. We think it can and should be reduced by 25-50% with careful editing and rethinking about what really needs to be discussed. The modeling parts of the Results and Discussion (see also point #4 below), in particular, need serious trimming and prioritization about what really needs to be discussed.

3) It is true that G_SH_ was not evident in the previous paper cited for the R1H mutant, but in that work they only show data down to -10 mV, a voltage where the shuttle current would be very small. That study focused on looking at accessibility of inserted His to Zn, not on explicitly looking for a shuttle conductance. One sentence in the Results section would suffice to accurately state that G_SH_ was not reported in earlier work, but that those authors did not pulse negative enough to see it.

4) We have serious reservations about the analysis of the modeling. First, the description of the generation of Hv1 E is not clear: "Representations of Hv1 E represent the first snapshot in a trajectory of ten possible structural conformations collected from a 1 ns MD production run." What is meant by "ten possible structural conformations"? Second, the analysis presented is based on single snapshots from MD simulations, which may or may not be representative, or may be one of multiple populated configurations, as the authors themselves point out in the last paragraph of the Discussion: "Because Arg side chains may exhibit a high degree of dynamic flexibility (9), distance measurements inferred from static snapshots here and elsewhere should be regarded as relatively coarse descriptions of conformational rearrangements in a highly dynamic protein domain." If only single snapshots are being reported, then what is the point of running MD simulations in the first place? And what is the criterion for a particular choice of snapshot? Were the trajectories clustered and the most populated cluster chosen? Third, with MD trajectories in hand, the authors have the opportunity to analyze internal hydration structure and dynamics, as well as electrostatic potential, as many others have done previously for Kv and Hv channels. However, no such analysis was performed, and statements such as the following (among others), are not substantiated by simple inspection of a single snapshot lacking water molecules:

"Hv1 D and E model structures are consistent with the notion that both gating charges and permeating protons cross the focused electrical field through a hydrated crevice."

"the central crevice remains substantially open, and may accommodate water molecules that are necessary for H^+^ shuttling via R1H"

"we find that the imidazole ring of R1H is apparently located within the solvent-accessible VS domain central crevice"

"Although Hv1 D and Hv1 E appear to be fully compatible with G_SH_ experimental data reported here, other putative resting-state Hv1 VS domain structures evidently do not contain sufficiently hydrated central crevices to accommodate H^+^-shuttle function."

"Taken together, the experimental data and model structures reported here suggest that G_AQ_ and G_SH_ share a common H^+^ permeation pathway with S4 gating charges and that translation of S4 during VS activation must be of sufficient amplitude to 'unblock' the hydrated central crevice to open G_AQ."

In short, we think that many of the conclusions emerging from the modeling are not substantiated by the data presented. We therefore request that the amount of space (text and figures) devoted to the modeling be greatly reduced. To the extent that you wish to draw conclusions from the modeling, we request that you quantitatively analyze your Hv1 models to provide more information on hydration and electrostatics.

---

## [Author Response]

Essential revisions:

*1) Throughout the manuscript the authors complicate the story by assuming that the voltage-dependence for G_SH_ must reflect a conformational change in the protein, rather than arising from an intrinsic rectification to proton permeation. Until we get to Figure 4, all the data actually would support a mechanism involving rectification of permeation. Also, the apparent shifts in the dG/dV analysis are only a hint. We suggest removing all verbiage about gating mechanisms surrounding G_SH_ until you get to the section of the Results where you wish to present the Figure 4 experiment. Then you can raise the issue about mechanism of voltage-dependence to G_SH_, remind the reader about mutants affecting G_AQ_ and not G_SH_, and then show how pH seems to shift both and talk about what that might mean. The authors ideas may be on the right track, but they don't have very firm ground to stand on and the presentation should reflect that.*

We appreciate the Reviewers’ keen insight and agree that it is difficult to ascertain with certainty whether G_SH_ gating results from large-scale movements of the protein (such as voltage-dependent translation of the S4 helix) vs. subtler, possibly local, conformational rearrangements (that might be more typical of pore block by a protein-associated moiety or ion). As suggested by the Reviewers, we have tried to remove potentially confusing verbiage about the mechanism of G_SH_ gating from the Results and to more carefully cite support for our interpretations of the data. We added a section to the Discussion addressing how voltage-dependent G_SH_ gating could result from local reorientation of the introduced His side chain within the electric field that is sharply focused across the gating pore.

We agree with the Reviewers’ concerns about the reported pH-dependent shifts in dG_STEP_/dV relations, and realize that even our most diligent efforts to quantify the pH_O_ dependence of G_SH_ gating are not entirely satisfying. We have tried numerous different experimental strategies to address this issue and present only the results that most compellingly argue that G_SH_ gating is modified by changes in pH. Other experimental and analytical approaches that are not presented here also yield tantalizingly believable evidence of pH-dependent G_SH_ gating, but ultimately are not more convincing than the reported V_PEAK_ shifts of dG_STEP_/dV relations. Given that the mechanism of pH-sensitive gating in Hv1 remains essentially unknown, we feel that this work makes a potentially important contribution to the field.

Finally, the results reported here (in which we reproducibly measure whole-cell proton currents to -200 mV in cells that express a cytotoxic, leaky H^+^ channel) represent a heroic effort on the part of the graduate student who conducted the majority of the experiments. We and others may find ways to improve our experimental approaches in the future, but the reported results represent our best effort so far to characterize G_SH_ gating under less-than-optimal experimental conditions.

*2) The paper is straightforward to read and understand for experts, but it is too longwinded and will be impenetrable for the outsider. Terms like Ftail and Fhook are thrown about and won't be clear to those who haven't read the Larsson paper. In its present form it will probably only be read by scientists working on Hv1 or those who are total nerds and simply love the topic of voltage sensing. We think it can and should be reduced by 25-50% with careful editing and rethinking about what really needs to be discussed. The modeling parts of the Results and Discussion (see also point #4 below), in particular, need serious trimming and prioritization about what really needs to be discussed.*

We have comprehensively revised the manuscript, eliminating non-essential verbiage and Figures. The revised Introduction and Results and Discussion now totals 8721 words (37% shorter than the originally-submitted 13,869 word manuscript).

*3) It is true that G_SH_ was not evident in the previous paper cited for the R1H mutant, but in that work they only show data down to -10 mV, a voltage where the shuttle current would be very small. That study focused on looking at accessibility of inserted His to Zn, not on explicitly looking for a shuttle conductance. One sentence in the Results section would suffice to accurately state that G_SH_ was not reported in earlier work, but that those authors did not pulse negative enough to see it.*

We have modified our treatment of the discrepancy between our results and the previous report by Kulleperuma, et al. (2013) in the Discussion:

“Our results are consistent with previous reports describing G_SH_ in other VS domain R1H mutants (1-3) but contrast with a previous study in Hv1 (4). One possible explanation for the discrepancy is that G_SH_ is difficult to measure when mutant Hv1 channel expression levels are low, as in the previous study (4), whereas the inducible expression system used here drives the high expression that is evidently necessary to reproducibly measure G_SH_.”

To clarify, Kulleperuma, et al. (2013) state in Results that “All three constructs were nonconducting at negative voltages and opened with time during depolarizing pulses. Specifically, we did not observe evidence for H^+^ conductance activated at negative voltages, or for proton carrier activity at intermediate voltages, as might have been expected by analogy with the His scanning studies of Starace and Bezanilla in *Shaker* K^+^ channels (Starace et al., 1997; Starace and Bezanilla, 2001, 2004).”

Furthermore, in their Conclusion the same authors state: “In summary, the only definitive conclusion from the properties of the Arg→His mutants of hHV1 is that R205H does not act as a hyperpolarization-activated proton channel at negative voltages.”

Our reading of the literature suggests that DeCoursey and colleagues were well aware of the experimental conditions necessary for G_SH_ measurement, but nevertheless arrived at an erroneous conclusion, and we are obliged to correct this factual error in the scientific record.

*4) We have serious reservations about the analysis of the modeling. First, the description of the generation of Hv1 E is not clear: "Representations of Hv1 E represent the first snapshot in a trajectory of ten possible structural conformations collected from a 1 ns MD production run." What is meant by "ten possible structural conformations"?*

We regret that the description of the Hv1 E simulation was unclear. We have modified the Methods to more clearly describe our in silico approaches and added additional information from MD trajectories to reinforce the veracity of our interpretations (see Figure 5—figure supplement 1, 3 and Figure 6—figure supplement 1). Part of the Reviewers’ concern could be a result of the fact that the Hv1 B and D models were originally produced and simulated in the UK, but Hv1 B N4R and Hv1 E (R1H) mutant models were subsequently built and subjected to MD simulation in the US. The fact that in silico approaches utilized by different individual researchers are not identical, but nonetheless yield highly similar structures, should be reassuring. In addition to new quantitative measurements of model structure behavior during MD simulations added to Figure supplements, we have clarified the Homology modeling and MD simulation section in Methods.

*Second, the analysis presented is based on single snapshots from MD simulations, which may or may not be representative, or may be one of multiple populated configurations, as the authors themselves point out in the last paragraph of the Discussion: "Because Arg side chains may exhibit a high degree of dynamic flexibility (9), distance measurements inferred from static snapshots here and elsewhere should be regarded as relatively coarse descriptions of conformational rearrangements in a highly dynamic protein domain." If only single snapshots are being reported, then what is the point of running MD simulations in the first place? And what is the criterion for a particular choice of snapshot? Were the trajectories clustered and the most populated cluster chosen?*

We realize that the statement about Arg side chain flexibility is confusing, and have revised this section for clarity.

“The apparent flexibility of Arg side chains in VS domains (5) suggests that the magnitude of S4 translation may not be easily inferred from measurements of gating valence alone. Proton transfer via G_SH_ and voltage-dependent block of G_AQ_ appear to place stringent constraints on the relative positions of target side chain atoms, and may offer advantages over alternative approaches, such as chemical accessibility in Cys mutant proteins, for ascertaining structural changes that occur during VS activation.”

Typical MD trajectories contain thousands to millions of individual frames, so choices must be made about which snapshots to use for static images in the published Figures. We have tried to select representative snapshots, but in fact each frame is in some way different from all others, and no snapshot is totally representative. We have added additional data from Hv1 D and Hv1 E MD trajectories (Figure 5—figure supplement 1 and Figure 5—figure supplement 3; Figure 6—figure supplement 1) showing that C_α_ atom distances and electrostatic interactions are constant, indicating that the structures are stable in the simulation environment. This data argues that distance measurements made between sites in the protein backbone are likely to be quite similar, regardless of which particular snapshot is used for comparison to other structures.

Are MD simulations necessary for this work? As the title of the manuscript indicates, our intention to use experimental data to constrain *possible* Hv1 models. MD simulations are necessary for ascertaining whether a particular model structure is thermodynamically stable, and represent a necessary quality check for model structures. The model structures presented here represent the culmination of many iterative attempts to build models that are most consistent with all of the available experimental data, but even our best efforts do not guarantee that any model will remain stable in the all-atom MD simulation environment after model-building, lipid equilibration, energy minimization and harmonic constraint.

MD simulations should not be interpreted as evidence that a given model is any more representative of a physiologically meaningful conformation, or that our preferred model is necessarily more accurate than any of myriad other possible models. Our experience indicates that software packages such as Modeller afford the investigator tremendous latitude at the model-building stage, and potentially important features of candidate model structures are subject to bias resulting from both the selection of different technical parameters and investigator ignorance or preference. We are encouraged by the observations that the models presented here are both stable in an MD environment (and thus energetically feasible) and are evidently superior to models built by other groups in that they are uniquely consistent with experimental data. We feel that atomic models which are validated by directed experiments offer more information content than less rigorously-tested models and X-ray structures determined from proteins subjected to biochemical torture, reconstitution, and crystallization under non-physiological conditions. We are hopeful that our sustained effort to successfully merge experimental and computational approaches will yield a deeper understanding of the structural basis of VS activation and H^+^ permeation mechanisms in Hv1.

*Third, with MD trajectories in hand, the authors have the opportunity to analyze internal hydration structure and dynamics, as well as electrostatic potential, as many others have done previously for Kv and Hv channels. However, no such analysis was performed, and statements such as the following (among others), are not substantiated by simple inspection of a single snapshot lacking water molecules:*

We previously reported that activated-state Hv1 model structures contain a well-hydrated central crevice (Ramsey et al., 2010) and are thus aware of the capabilities and limitations or water modeling in standard MD simulations. However, we omitted descriptions of hydration and water structure in resting-state Hv1 models here because the analysis did not seem to add substantially to the experimental results, which strongly argue that the central crevice is hydrated. Indeed, most of the points of concern raised below are statements that are based on experimental observations, not model structures. However, to allay the concerns of the Reviewers, we added Figures showing central crevice water occupancy in Hv1 D and Hv1 E resting-state models during MD simulations. We have also substantially modified the text as suggested, and hope that the revised manuscript fairly balances experimental constraints and structural inferences to yield innovates mechanistic hypotheses.

*"Hv1 D and E model structures are consistent with the notion that both gating charges and permeating protons cross the focused electrical field through a hydrated crevice."*

We believe that this statement is well supported by available evidence and not based on observations of models or their behavior in MD simulations. We clearly state that our new model structures are merely consistent with multiple experimental observations here and elsewhere showing that a single residue substitution (R1H) is sufficient to confer G_SH_ (Starace, et al., 2004; Villalba-Galea, et al., 2013).

*"the central crevice remains substantially open, and may accommodate water molecules that are necessary for H^+^ shuttling via R1H"*

H^+^ shuttling via G_SH_ evidently requires water molecules to function as proton donor to and acceptor from the introduced R1 His (Starace, et al., 2004). Without water, the mechanism by which H^+^ would manage to cross the thickness of the membrane is difficult to conceive. We showed previously (Ramsey et al., 2010) that ionizable residues throughout Hv1 are dispensable for H^+^ conduction, so the possibility that a latent proton wire exists somewhere in the protein and is selectively enabled by R1H seems remote. Another possibility is that protons somehow tunnel through the protein, but we have no way to directly test this hypothesis using conventional experimental approaches. Given that a number of other studies (including Krepkiy et al., 2012) show or imply that VS domains contain substantially hydrated crevices, we infer water is necessary for H^+^ shuttling and that this is the simplest explanation for the experimental data.

"we find that the imidazole ring of R1H is apparently located within the solvent-accessible VS domain central crevice"

Newly added data in Figure 5—figure supplement 3 and 3I directly addresses this concern.

*"Although Hv1 D and Hv1 E appear to be fully compatible with G_SH_ experimental data reported here, other putative resting-state Hv1 VS domain structures evidently do not contain sufficiently hydrated central crevices to accommodate H^+^-shuttle function."*

This conclusion is drawn from our inspection of hydrophobic packing in *Ci* Hv1, mHv1cc and Hv1FL model and X-ray structures (Figure 6 and Figure 6—figure supplement 1), which do not contain sufficient space to accommodate water molecules that could function to bridge R1H to the intra- and extra-cellular bulk solvent. Hv1 D and Hv1 E are clearly distinct from *Ci* Hv1, mHv1cc and Hv1FL in this regard, and we have added additional Figures to support this interpretation (Figure 5—figure supplement 3).

*"Taken together, the experimental data and model structures reported here suggest that G_AQ_ and G_SH_ share a common H^+^ permeation pathway with S4 gating charges and that translation of S4 during VS activation must be of sufficient amplitude to 'unblock' the hydrated central crevice to open G_AQ."*

Although we think that there is ample experimental evidence in the manuscript and literature to support this conclusion, we modified the text and added new data to address the Reviewers’ concern. Our interpretation that the S4 gating charges block the H^+^ permeation pathway is based on the results of previous studies of G_Ω_ and G_SH_ in Shaker and other channels (6, 7); to this literature we add the observation that N4R creates a voltage-dependent block of G_AQ_. X-ray and model structures strongly suggest that the central crevice (‘gating pore’) is the only continuously hydrated pathway in VS domains, and we and others agree that the simplest explanation is that traversing H^+^ and gating charges share the ‘gating pore’ (1, 6-10). This possibility is further supported by our MD simulations, in which we observe robust hydration of the central crevice (see new panels in Figure 5—figure supplement 3). Thus, experimental data and structural studies appear to mutually support the hypothesis that proton flux through G_AQ_ and G_SH_ utilize the VS ‘gating pore’ for H^+^ transfer in Hv1.

*In short, we think that many of the conclusions emerging from the modeling are not substantiated by the data presented. We therefore request that the amount of space (text and figures) devoted to the modeling be greatly reduced. To the extent that you wish to draw conclusions from the modeling, we request that you quantitatively analyze your Hv1 models to provide more information on hydration and electrostatics.*

To address the Reviewers’ concerns we have: 1) Reduced the number of main Figures showing structural representations to two (Figure 5 and Figure 6) and moved important (but visually dense) comparisons between Hv1 model and X-ray structures to the Figure supplements; 2) added data from MD simulations (Figure 5—figure supplement 1,Figure 5—figure supplement 3 and Figure 6—figure supplement 1) that supports our conclusions about structure and water accessibility in Hv1 B N4R, Hv1 D and Hv1 E; 3) added a comparison to the *At* TPC1 DII VS domain X-ray structure (Figure 5); and 4) substantially reduced the amount of text in the Results and Conclusions sections dealing with structural comparisons.

Our conclusions about central crevice hydration, water access to R1H, G_AQ_ block by S4 Arg gating charges, and gating pore/proton transfer pathways utilizing the same structural feature are mainly based on experimental data, not on modeling or MD simulations. We reiterate that one of the main intentions of this manuscript is to demonstrate how carefully designed and executed experiments can be used to constrain structural conformations of the VS domain. We feel that the manuscript has the potential to make a valuable contribution to the field by comparing and contrasting our new Hv1 models with previously published model and X-ray structures, most of which do not satisfy the expectations imposed by G_SH_ and G_AQ_ experimental data in some important way.